# HybridNorm: Towards Stable and Efficient Transformer Training via Hybrid Normalization

**Zhijian Zhuo**[1, 2]   **Yutao Zeng**[2, †]   **Ya Wang**[2, †]   **Sijun Zhang**[2]   **Jian Yang**[3]
**Xiaoqing Li**[4]   **Xun Zhou**[2]   **Jinwen Ma**[1]

[1] School of Mathematical Sciences, Peking University   [2] ByteDance Seed
[3] Beihang University   [4] Capital University of Economics and Business

## Abstract

Transformers have become the de facto architecture for a wide range of machine learning tasks, particularly in large language models (LLMs). Despite their remarkable performance, many challenges remain in training deep transformer networks, especially regarding the position of the layer normalization. While Pre-Norm structures facilitate more stable training owing to their stronger identity path, they often lead to suboptimal performance compared to Post-Norm. In this paper, we propose **HybridNorm**, a simple yet effective hybrid normalization strategy that integrates the advantages of both Pre-Norm and Post-Norm. Specifically, HybridNorm employs QKV normalization within the attention mechanism and Post-Norm in the feed-forward network (FFN) of each transformer block. We provide both theoretical insights and empirical evidence to demonstrate that HybridNorm improves the gradient flow and the model robustness. Extensive experiments on large-scale transformer models, including both dense and sparse variants, show that HybridNorm consistently outperforms both Pre-Norm and Post-Norm approaches across multiple benchmarks. These findings highlight the potential of HybridNorm as a more stable and effective technique for improving the training and performance of deep transformer models. Code is available at `https://github.com/BryceZhuo/HybridNorm`.

## 1   Introduction

Transformers have become the backbone of large language models (LLMs) and a wide range of machine learning applications. These architectures are capable of modeling long-range dependencies through self-attention mechanisms, which have made them the preferred choice for a variety of tasks, including language modeling, machine translation, and image processing [1–3]. However, as transformer models become deeper and more complex, ensuring stable training remains a significant challenge. One critical factor that influences training stability is the choice of normalization methods, which is crucial for mitigating issues such as internal covariate shift and gradient instability [4]. Effectively addressing these challenges is crucial for fully harnessing the potential of deep transformer models in large-scale applications.

In transformers, Layer Normalization (LayerNorm) [5] plays a central role in stabilizing training by normalizing the activations within each layer. The two predominant strategies for applying LayerNorm are Pre-Layer Normalization (Pre-Norm) and Post-Layer Normalization (Post-Norm), each with its respective benefits and trade-offs. In the Pre-Norm architecture, normalization is

---

[†]Corresponding author: Yutao Zeng (yutao.zeng@outlook.com) and Ya Wang (wangyazg@gmail.com).

39th Conference on Neural Information Processing Systems (NeurIPS 2025).

applied before the residual addition, resulting in a more prominent identity path that facilitates faster convergence and more stable gradients [4]. This design is particularly advantageous when training deep models, as it helps mitigate gradient-related issues that can arise during backpropagation. However, while Pre-Norm can stabilize training, it often leads to inferior final performance compared to Post-Norm [6]. In contrast, Post-Norm applies normalization after the residual connection, resulting in stronger regularization effects, which contribute to improved model performance. This approach has been shown to improve the generalization ability of transformers, particularly in very deep networks [7]. Further discussion of related work is provided in Appendix A.

Despite the benefits of each approach, there is an inherent trade-off between training stability and final model performance. Pre-Norm structures typically stabilize training but may underperform in terms of generalization, while Post-Norm architectures provide better performance but can be more difficult to train, especially in deep models. To reconcile these trade-offs, we propose a hybrid normalization method that applies QKV normalization in the attention mechanism and Post-Norm in the feed-forward network (FFN), which is named as HybridNorm. The QKV normalization in the attention mechanism stabilizes the flow of information between layers by normalizing the query, key, and value components, while Post-Norm in the FFN ensures the effective depth of the transformer.

Through extensive experiments on large-scale models, we validate the effectiveness of our approach. Our results show that the hybrid normalization method significantly outperforms both Pre-Norm and Post-Norm across multiple benchmarks, providing a stable training process and improved model performance. We believe that this hybrid approach offers a promising solution for enhancing the training stability and performance of deep transformer architectures, particularly in the rapidly evolving field of LLMs. The main contributions of this paper can be summarized as follows:

- We propose HybridNorm, a novel hybrid normalization structure that combines the advantages of Pre-Norm and Post-Norm, offering a simple yet effective solution to enhancing performance in large transformer models. Our method is designed to exploit the strengths of both normalization approaches, ensuring robust convergence during training and superior final performance.

- We present both theoretical and empirical analyses of HybridNorm, demonstrating its advantages in enhancing gradient flow stability and improving model robustness. Our findings underscore the method's effectiveness in mitigating core challenges inherent to deep transformer architectures.

- Through extensive experiments on large-scale models (550M - 7B), we empirically validate the effectiveness of our approach. Our results show that hybrid normalization significantly outperforms both Pre-Norm and Post-Norm across a variety of tasks, leading to more stable training and improved model performance, particularly in the context of LLMs.

## 2 Preliminaries

**Scaled Dot-Product Attention**  The scaled dot-product attention computes the attention scores between the Query (Q) and Key (K) matrices, scaled by the square root of the key dimension $d_k$, and applies these scores to the Value (V) matrix. The formulation is expressed as

$$\text{attn}(Q, K, V) = \text{softmax}\left(\frac{QK^\top}{\sqrt{d_k}}\right) V, \tag{1}$$

where $Q, K, V \in \mathbb{R}^{n \times d_k}$ represent the query, key, and value matrices respectively, and $n$ is the sequence length.

**Multi-Head Attention**  Multi-head attention (MHA) extends the scaled dot-product attention mechanism by splitting the query, key, and value matrices into $h$ heads, each of size $d_k = d/h$. Each head independently computes attention scores, and the outputs are concatenated and linearly projected to the original dimension,

$$\text{MHA}(X) = \text{Concat}(\text{head}_1, \ldots, \text{head}_h)W^O, \tag{2}$$

where $\text{head}_i = \text{attn}(Q_i, K_i, V_i)$ for $i = 1, 2, \ldots, h$, $\{\bullet_i\}_{i=1}^h = \text{Split}(XW_\bullet)$ for $\bullet \in \{Q, K, V\}$, and $W_Q, W_K, W_V, W_O \in \mathbb{R}^{d \times d}$ are learnable parameters. By enabling the model to focus on different subspaces of the input representation, MHA enhances the transformer's capacity to capture diverse patterns in the input sequence.

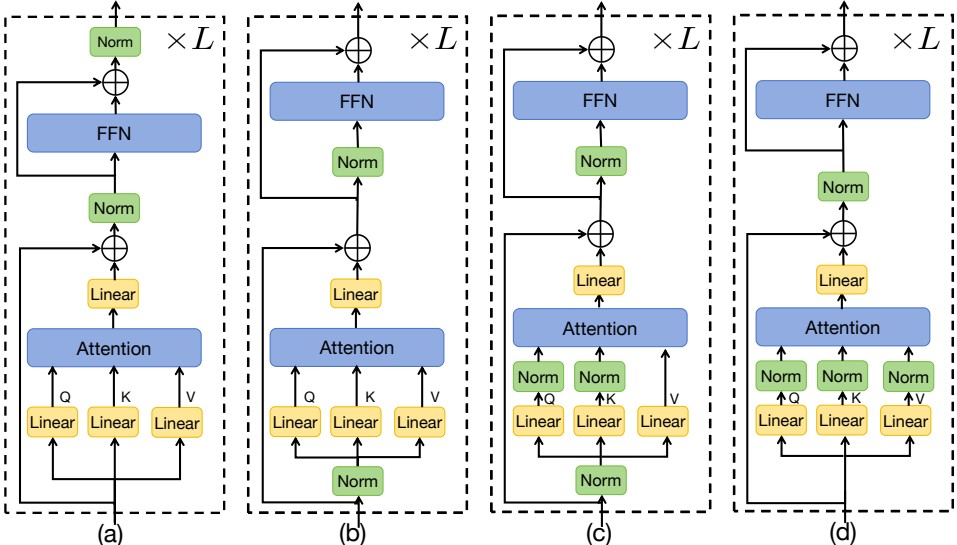

Figure 1: Illustrations of different transformer layer structures: (a) Post-Norm architecture; (b) Pre-Norm architecture; (c) Pre-Norm with QK-Norm architecture; (d) HybridNorm architecture.

## 2.1 Post-Norm and Pre-Norm

The transformer architecture is composed of a stack of L blocks, each consisting of two key components: MHA and FFN. Residual connections and normalization layers are applied around both the MHA and FFN in each block to facilitate effective training and improve model stability. Figure 1 (a)&(b) illustrate Post-Norm and Pre-Norm, respectively.

**Post-Norm**  Post-Norm applies the normalization layer after the residual connection in each transformer sub-layer. Formally, the output of Post-Norm can be expressed as

$$Y^l = \text{Norm}(\text{MHA}(X^l) + X^l), \quad X^{l+1} = \text{Norm}(\text{FFN}(Y^l) + Y^l), \tag{3}$$

where $\text{Norm}$ denotes RMSNorm [8] or LayerNorm [5].

**Pre-Norm**  In contrast, Pre-Norm normalizes the input to the sub-layer, which allows for a more prominent identity path. The output of Pre-Norm is given by

$$Y^l = \text{MHA}(\text{Norm}(X^l)) + X^l, \quad X^{l+1} = \text{FFN}(\text{Norm}(Y^l)) + Y^l. \tag{4}$$

This structure facilitates better gradient flow and stable convergence, particularly for deep models. However, its reliance on normalization before the residual connection can lead to suboptimal performance compared to Post-Norm, as the normalization does not account for the interaction between the residual connection and the sub-layers output. An analysis of the fundamental differences between the two approaches is provided in the Appendix F.

## 3  HybridNorm

To address the trade-offs between Post-Norm and Pre-Norm, we propose **HybridNorm**, a hybrid normalization strategy that integrates their strengths. Specifically, HybridNorm combines **QKV-Norm** [9, 10] in MHA and **Post-Norm** in FFN.

**QKV Normalization in Attention**  In the attention mechanism, the query, key, and value matrices are normalized individually before computing the attention output. The normalized QKV matrices are then used in the scaled dot-product attention. QKV-Norm enhances the stability of model training

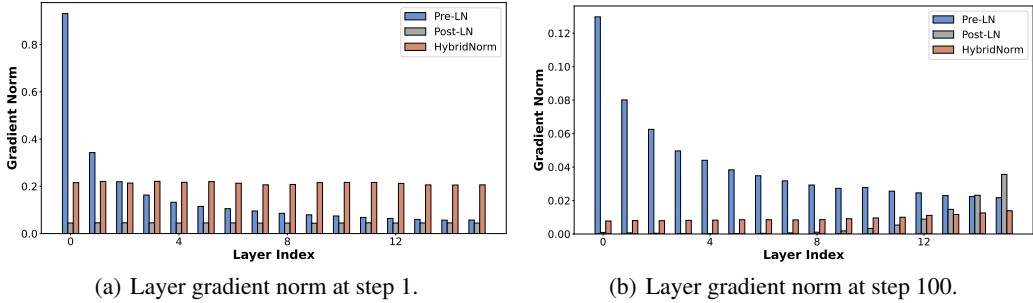

(a) Layer gradient norm at step 1.    (b) Layer gradient norm at step 100.

Figure 2: Gradient norm of Pre-Norm, Post-Norm, and HybridNorm at different training steps.

and leads to improved downstream performance. Formally, attention with QKV-Norm is defined as

$$\text{attn}_{QKV}(Q, K, V) = \text{softmax}\left(\frac{\text{Norm}(Q)\text{Norm}(K)^{\top}}{\sqrt{d_k}}\right)\text{Norm}(V). \tag{5}$$

And we denote the multi-head attention with $\text{attn}_{QKV}$ as $\text{MHA}_{QKV}$.

**HybridNorm Architecture**    Combining the above, the overall output of a transformer block with HybridNorm can be expressed as

$$Y^l = \text{MHA}_{QKV}(X^l) + X^l, \quad X^{l+1} = \text{FFN}(\text{Norm}(Y^l)) + \text{Norm}(Y^l). \tag{6}$$

The architecture illustration can be found in Figure 1(d) and the pseudocode is shown in Algorithm 1. By integrating QKV normalization in the attention mechanism and Post-Norm in the FFN, HybridNorm achieves stable training dynamics and enhanced final performance. The theoretical gradient analysis can be found in Appendix B.

**Remark 1.** *The method most closely related to ours is Mix-LN [11], which applies Post-Norm to the earlier layers and Pre-Norm to the deeper layers, resulting in enhanced training stability and performance. In contrast, our proposed HybridNorm integrates Pre-Norm and Post-Norm within each transformer layer, thereby providing a more uniform approach across different layers to leverage the benefits of both normalization strategies. Moreover, experiments demonstrate that HybridNorm achieves superior downstream performance compared to Mix-LN (see Table 6 & Table 12).*

**Special Treatment of First Block**    Inspired by prior work [12], which employs the Mixture of Experts (MoE) architecture with specialized handling of the first layer, we explore the impact of introducing specialized normalization to the first transformer block. In our approach, the first layer of the transformer is treated differently by applying Pre-Norm on MHA and FFN, while maintaining QKV-Norm. Specifically, the structure of our first layer is defined as

$$Y^0 = \text{MHA}_{QKV}(\text{Norm}(X^0)) + X^0, \quad X^1 = \text{FFN}(\text{Norm}(Y^0)) + Y^0. \tag{7}$$

We refer to this variation of HybridNorm, which incorporates the specialized first block treatment, as HybridNorm$^*$. This design aims to stabilize the training of the first transformer block and boost overall performance by improving the flow of gradients in the early stages of training.

## 4    Theoretical Analysis

### 4.1    Benefits of Hybrid Method

To gain deeper insights into the stability introduced by HybridNorm, we follow the approach of [13, 11] and analyze the evolution of gradient norms throughout training iterations. Suppose $x$ and $F$ are the input and the sublayer of Transformer, respectively. The output of Post-Norm is $y_{Post} = \text{Norm}(x + F(x))$ and the output of Pre-Norm is $y_{Pre} = x + F(\text{Norm}(x))$. Then we have

$$\frac{\partial y_{Post}}{\partial x} = \frac{\partial \text{Norm}(x + F(x))}{\partial(x + F(x))}\left(I + \frac{\partial F(x)}{\partial x}\right), \tag{8}$$

$$\frac{\partial y_{Pre}}{\partial x} = I + \frac{\partial F(\mathrm{Norm}(x))}{\partial \mathrm{Norm}(x)} \frac{\partial \mathrm{Norm}(x)}{\partial x}, \tag{9}$$

where the gradient of normalization is $\frac{\partial \mathrm{Norm}(x)}{\partial x} = \alpha \odot \frac{\sqrt{d}}{||x||_2} \left( I - \frac{xx^\top}{||x||_2^2} \right)$. The gradient of Post-Norm is the product of two gradients, one of which is the normalization. If the spectral radius of the normalization gradient is less than 1, it causes gradient vanishing. In Pre-Norm, the residual connection is isolated from the normalization, ensuring that gradients retain a lower bound, thereby preventing gradient vanishing. To ensure relatively stable gradients, a natural idea is to use both types of normalization within a Transformer block, leading to Pre-Post (Pre-Norm in MHA and Post-Norm in FFN) and Post-Pre. From Table 6, we find that Pre-Post achieves the best performance, which is why we adopt this. And placing Pre-Norm in MHA with QKV-Norm further enhances performance.

In Figure 2, we compare the gradient norms of Pre-Norm, Post-Norm, and HybridNorm at steps 1 and 100. The results indicate that Pre-Norm tends to exhibit gradient explosion in deeper models, while Post-Norm suffers from vanishing gradients, both of which hinder effective optimization. In contrast, HybridNorm maintains a well-balanced gradient flow throughout training, effectively mitigating these issues. An intuitive understanding is that Pre-Norm tends to amplify gradients, while Post-Norm diminishes them. HybridNorm alternates between these two normalization strategies, leading to more stable gradient propagation during backpropagation and effectively preventing gradient explosion or vanishing. This balanced gradient propagation contributes to smoother optimization dynamics and faster convergence, further reinforcing the effectiveness of HybridNorm in stabilizing training.

## 4.2 Benefits of QKV-Norm

Theoretically, we study how QKV-Norm affects the gradient flow during backpropagation, which is crucial for training stability in deep transformer models. Our analysis reveals that QKV-Norm helps decouple gradients between different weight matrices, thereby stabilizing training.

**Theorem 1** (Informal version of Theorem 2). *Suppose the the output of the attention is $S$, the input $X \in \mathbb{R}^{s \times d}$, parameters $W_Q, W_K, W_V, W_O^\top \in \mathbb{R}^{d \times d_k}$. For the attention with **Pre-Norm**, we have*

$$\left\| \frac{\partial S}{\partial W_O} \right\|_F = \mathcal{O}\left( \|W_V\|_2 \right), \left\| \frac{\partial S}{\partial W_V} \right\|_F = \mathcal{O}\left( \|W_O\|_F \right),$$

$$\left\| \frac{\partial S}{\partial W_Q} \right\|_F = \mathcal{O}\left( \|W_K\|_2 \|W_V\|_2 \|W_O\|_2 \right), \left\| \frac{\partial S}{\partial W_K} \right\|_F = \mathcal{O}\left( \|W_Q\|_2 \|W_V\|_2 \|W_O\|_2 \right).$$

*For the attention with **Pre-Norm and QK-Norm**, we have*

$$\left\| \frac{\partial S}{\partial W_O} \right\|_F = \mathcal{O}(\|W_V\|_2), \left\| \frac{\partial S}{\partial W_V} \right\|_F = \mathcal{O}(\|W_O\|_F), \left\| \frac{\partial S}{\partial W_Q} \right\|_F = \left\| \frac{\partial S}{\partial W_K} \right\|_F = \mathcal{O}(\|W_V\|_2 \|W_O\|_2).$$

*For the attention with **QKV-Norm**, we have*

$$\left\| \frac{\partial S}{\partial W_O} \right\|_F = \mathcal{O}(1), \left\| \frac{\partial S}{\partial W_V} \right\|_F = \mathcal{O}(\|W_O\|_2), \left\| \frac{\partial S}{\partial W_Q} \right\|_F = \left\| \frac{\partial S}{\partial W_K} \right\|_F = \mathcal{O}(\|W_O\|_2).$$

The above theorem is an informal version of Theorem 2, with a more precise statement and proof provided in Appendix B. In attention with Pre-Norm, gradients of weights exhibit strong dependencies on other weights; for instance, $W_Q$ and $W_K$ are influenced by all three other weights but not by themselves. In contrast, with QKV-Norm, the gradient of each weight depends at most on itself and $W_O$. This indicates that the gradient in Pre-Norm is more tightly coupled with other weights compared to QKV-Norm, while Pre-Norm with QK-Norm lies between the two. Consequently, during training, if the norm of a certain weight becomes excessively large, it is harder to control in Pre-Norm, leading to increased gradient magnitude. This creates a vicious cycle that may cause model collapse. In contrast, QKV-Norm alleviates this issue and significantly improves training stability. In summary, the degree of gradient coupling follows: Pre-Norm > Pre-Norm with QK-Norm > QKV-Norm; whereas training stability follows the reverse: Pre-Norm < Pre-Norm with QK-Norm < QKV-Norm.

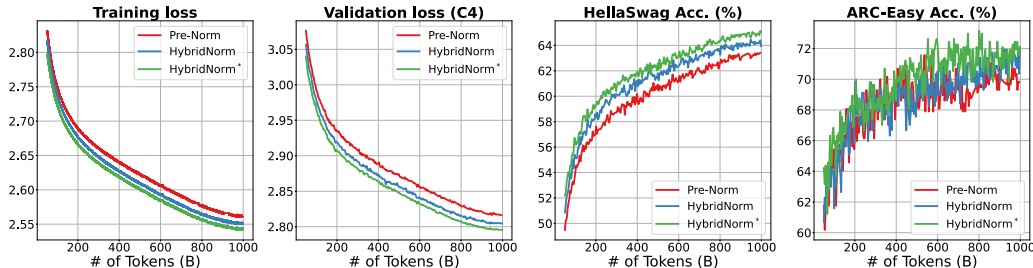

Figure 3: Training dynamics for 1.2B dense models with Pre-Norm, HybridNorm and HybridNorm*
under 1T training tokens. We present the training loss, validation loss, and downstream performance
on HellaSwag and ARC-Easy, demonstrating that HybridNorm* achieves superior performance.

Table 1: Downstream evaluation results of 1.2B dense models with Pre-Norm, HybridNorm, and
HybridNorm* under 1T training tokens. OQA refers to OpenbookQA.

| Methods | BasicArithmetic | HellaSwag | SciQ | ARC-C | ARC-E | PIQA | OQA | COPA | Avg.↑ |
|---|---|---|---|---|---|---|---|---|---|
| Pre-Norm | 44.10 | 63.41 | 91.80 | **39.20** | 69.82 | 75.19 | 38.40 | 82.00 | 62.99 |
| HybridNorm | 44.12 | 64.22 | **91.88** | 39.13 | 71.05 | 74.72 | 38.88 | 82.00 | 63.25 |
| HybridNorm* | **47.21** | **65.12** | 91.38 | 37.06 | **71.79** | **75.72** | **39.16** | **85.78** | **64.15** |

## 5 Experiments

### 5.1 Experiment Settings

**Baseline**    We evaluate HybridNorm across two series of models: dense models and Mixture of
Experts (MoE) models. The dense models include two scales: 550M and 1B, with the latter containing
approximately 1.27 billion parameters and utilizing an architecture similar to LLaMA 3.2 [14]. All
analytical experiments are conducted on the 550M dense models. For the MoE model, we use
the OLMoE framework [15], which activates 1.3B parameters out of a total of 6.9B parameters
(MoE-1B-7B). Both models are trained from scratch on the OLMoE Mix dataset [15].

**Model Configuration**    The 550M dense model has a model dimension of 1536, an FFN dimension
of 4096, and utilizes 16 attention heads with 4 key/value heads per attention head. The 1.2B model
features a larger model dimension of 2048 and an FFN dimension of 9192, with 32 attention heads
and 8 key/value heads per attention head. The MoE-1B-7B model employs 16 attention heads, a
model dimension of 2048, and an FFN dimension of 1024. Notably, it features 8 experts out of 64,
providing a more fine-grained distribution of computational resources. All models consist of 16 layers
and are trained with a consistent context length of 4096. More details can be found in Appendix C.

**Hyperparameters**    Model weights are initialized using Megatron initialization [16] (See Section
5.4 for more details). For the optimization, we apply the AdamW optimizer with $\beta_1 = 0.9$ and
$\beta_2 = 0.95$. All models are trained on sequences of 4096 tokens. For the dense model, we set the
learning rate to 3e-4, decaying to 3e-5 using a cosine scheduler. The MoE model starts with a learning
rate of 4e-4, decaying with a cosine schedule. We summarize the hyperparameters in Table 10.

**Evaluation Metrics**    To evaluate the performance of LLMs with HybridNorm, we employ a
diverse set of open benchmarks, including ARC-Easy (ARC-E) [17], ARC-Challenge (ARC-C) [17],
HellaSwag [18], PIQA [19], SciQ [20], CoQA [21], Winogrande [22], MMLU [23], BoolQ [24],
COPA [25], CSQA [26], OBQA [27], and SocialIQA [28]. We leverage the LM Eval Harness [29]
for standardized performance evaluation.

### 5.2 Main Results

**Dense Models**    We evaluate the performance of HybridNorm and HybridNorm* on 1.2B dense trans-
former models. Figure 3 compares the training dynamics of dense models with different normalization
methods. As shown in the figure, models with HybridNorm and HybridNorm* exhibit consistently

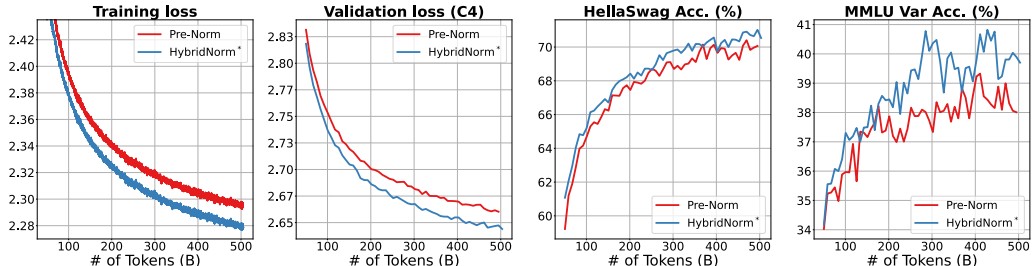

Figure 4: Training dynamics for MoE-1B-7B models with Pre-Norm and HybridNorm* under 500B training tokens. We present the training loss, validation loss, and downstream performance on HellaSwag and MMLU Var, demonstrating that HybridNorm* achieves superior performance.

Table 2: Downstream evaluation results of MoE-1B-7B with Pre-Norm and HybridNorm* under 500B training tokens. OQA refers to OpenbookQA.

| Methods | HellaSwag | ARC-C | ARC-E | PIQA | WinoGrande | OQA | BoolQ | COPA | Avg.↑ |
|---|---|---|---|---|---|---|---|---|---|
| Pre-Norm | 69.94 | 39.92 | 73.37 | 77.82 | 63.34 | 42.37 | 67.47 | 85.40 | 64.95 |
| HybridNorm* | **70.71** | **42.27** | **75.77** | **78.06** | **64.58** | **43.18** | **68.41** | **86.00** | **66.12** |

lower training loss and validation perplexity throughout training compared to Pre-Norm, highlighting their effectiveness in enhancing training stability and convergence. Table 1 presents the downstream evaluation results. HybridNorm* consistently outperforms Pre-Norm in most tasks, achieving the highest average score. Notably, it demonstrates substantial improvements in tasks such as BasicArithmetic (+3.11), HellaSwag (+1.71), and COPA (+3.78), indicating enhanced generalization and robustness. These results underscore the scalability of HybridNorm* in larger transformer models, further validating its effectiveness in improving both training stability and downstream performance. More results can be found in Figure 9. In Appendix E.1, we present additional comparisons with other approaches, such as Post-Norm and Mix-LN. We further provide analyses of signal propagation and entropy dynamics across different methods, as detailed in Appendix E.2 & E.3.

**MoE Models** For MoE models, we conduct experiments on MoE-1B-7B with 8 experts selected from a pool of 64. Figure 4 presents the training dynamics of MoE models under different normalization strategies. Throughout the training, HybridNorm* consistently achieves lower training loss and validation perplexity compared to Pre-Norm. These findings indicate that HybridNorm* effectively alleviates optimization difficulties in large-scale MoE models, resulting in more stable training and enhanced downstream performance. Further, as shown in Table 2, HybridNorm* consistently outperforms Pre-Norm across various downstream tasks, achieving the highest average score. Notably, it demonstrates significant improvements in ARC-C (+2.35), ARC-E (+2.40), and OpenbookQA (+0.81), highlighting its ability to enhance generalization across diverse benchmarks.

## 5.3 More Experiment in 7B Dense Model

The experimental setup primarily follows the 7B model configuration outlined in [30] and the number of training tokens is 150B. To further enhance model stability, we introduce an additional normalization layer to the output of the attention module in 7B model, with its weight initialized to $1/\sqrt{2L}$ [31], where $L$ denotes the number of model layers. Table 3 presents the training loss and validation perplexity (PPL), while Table 4 reports the downstream evaluation metrics.

The experimental results demonstrate the clear superiority of HybridNorm* over the traditional Pre-Norm approach in the 7B model. Firstly, HybridNorm* achieves a lower training loss (2.430 vs. 2.469), indicating more efficient optimization during training. This improvement in training dynamics translates into consistently better performance across a range of language modeling benchmarks. Specifically, HybridNorm* yields lower perplexity scores on all evaluated datasets, including C4, Books, Common Crawl, Wiki, and Wikitext 103. For instance, perplexity on the C4 dataset drops from 15.32 to 14.83, suggesting stronger generalization across both structured and unstructured corpora.

Table 3: Training loss and perplexity (PPL) comparison of Pre-Norm and HybridNorm* for 7B dense mdoel across multiple datasets. CC means Common Crawl.

| Methods | Loss | C4 | Books | CC | peS2o | Reddit | Stack | Wiki | Pile | Wikitext |
|---|---|---|---|---|---|---|---|---|---|---|
| Pre-Norm | 2.469 | 15.32 | 13.37 | 17.10 | 8.07 | 20.31 | 3.57 | 9.96 | 7.85 | 10.09 |
| HybridNorm* | **2.430** | **14.83** | **12.77** | **16.77** | **7.67** | **19.65** | **3.40** | **9.34** | **7.81** | **9.16** |

Table 4: Downstream evaluation results of 7B dense models trained with Pre-Norm and HybridNorm*. All numbers denote task accuracies (%). BA and OAQ mean Basic Arithmetic and Openbook QA, respectively.

| Methods | BA | Hella Swag | SciQ | ARC-C | ARC-E | PIQA | OQA | COPA | Wino Grande | BoolQ | Avg.↑ |
|---|---|---|---|---|---|---|---|---|---|---|---|
| Pre-Norm | 43.50 | 69.03 | 46.57 | 41.47 | 74.95 | 76.71 | 39.40 | 84.00 | 63.00 | 67.43 | 60.61 |
| HybridNorm* | **50.67** | **70.77** | **47.44** | **43.82** | **75.82** | **78.93** | **43.77** | **86.01** | **63.32** | **70.06** | **63.06** |

Moreover, HybridNorm* shows consistent improvements on all downstream tasks, covering various domains such as arithmetic reasoning, commonsense QA, and natural language inference. Notably, performance on Basic Arithmetic improves significantly from 43.50% to 50.67%. Across the full suite of tasks, including ARC, PIQA, COPA, BoolQ, and WinoGrande, HybridNorm* outperforms Pre-Norm in every case, leading to an overall increase in average accuracy from 60.61% to 63.06%.

## 5.4 Ablation Studies

**Initialization** To evaluate the sensitivity of Pre-Norm and HybridNorm to initialization schemes, we conduct ablation studies comparing three widely used initialization strategies: *Normal initialization* [32], *Depth-Scaled initialization* [33, 34], and *Megatron initialization* [16]. Normal initialization initializes all weights of linear layers using a truncated normal distribution with mean zero and standard deviation $1/\sqrt{2.5d}$, where $d$ is the hidden dimension. Depth-Scaled initialization and Megatron initialization introduce scaling factors to stabilize training in deep architectures. Specifically, Depth-Scaled initialization scales down the output projections of the attention and FFN by a factor of $\sqrt{2l}$, where $l$ is the layer index. In contrast, Megatron initialization scales down these projections by $\sqrt{2L}$, where $L$ is the total number of layers, mitigating gradient variance accumulation in very deep transformers. As shown in Table 5, Pre-Norm and HybridNorm exhibit sensitivity across different initialization methods, achieving the lowest training loss and perplexity under Normal initialization and Megatron initialization, respectively. Therefore, we set the default initialization method for Pre-Norm to Normal initialization and for HybridNorm to Megatron initialization in all experiments, respectively, which ensures that even under settings that may be more favorable to baseline models, the superiority of our approach is effectively demonstrated.

**Normalization Position** We investigate the impact of the position of normalization layers within the transformer block. **First**, we examine the effect of varying the placement of QKV normalization (e.g., normalization setting in attention). We extend the normalization setting by considering not only the Query (Q), Key (K), and Value (V) components but also the Context (C), which refers to the output of the attention mechanism. For instance, QKVC-Norm applies normalization to all four components: Query, Key, Value, and Context, while KV-Norm and KC-Norm focus on the normalization of the Key-Value and Key-Context pairs, respectively. QKVC-Post refers to transformer blocks that employ QKVC-Norm in the MHA while using Post-Norm in the FFN. **Second**, we explore the effect of integrating QKV-Norm into different transformer architectures. For instance, Pre-QKV-Post refers to a configuration where QKV-Norm is applied with Pre-Norm in the MHA layer, while the FFN layer utilizes Post-Norm. Other configurations follow similar definitions. **Finally**, we compare various hybrid combinations of Pre-Norm and Post-Norm. Pre-Post refers to transformer blocks that apply Pre-Norm in the MHA and Post-Norm in the FFN, whereas Post-Pre adopts the opposite configuration. Mathematical formulas for methods mentioned above can be found in Appendix G.

Table 5: Training loss and validation perplexity of 550M dense models with Pre-Norm and Hybrid-Norm under various initialization methods and 400B training tokens.

| Method | Initialization | Loss↓ | PPL on C4↓ | Method | Initialization | Loss↓ | PPL on C4↓ |
|--------|---------------|-------|------------|--------|---------------|-------|------------|
| | Normal | **2.75** | **20.29** | | Normal | 2.76 | 20.44 |
| Pre-Norm | Depth-Scaled | 2.76 | 20.49 | HybridNorm | Depth-Scaled | 2.76 | 20.40 |
| | Megatron | 2.76 | 20.44 | | Megatron | **2.74** | **20.00** |

Table 6: Abalation study of the position of normalization layers on 550M dense models with 400B tokens. We report the training loss, the perplexity on C4 and Pile, and the accuracy on HS (HellaSwag).

| Methods | Loss↓ | C4↓ | Pile↓ | HS↑ |
|---------|-------|-----|-------|-----|
| QKVC-Post | 2.74 | 20.05 | 10.34 | 52.68 |
| QKC-Post | 2.73 | 20.00 | 10.31 | 52.26 |
| QK-Post | - | | diverge | |
| KV-Post | 2.74 | 20.11 | 10.38 | 52.10 |
| KC-Post | 2.75 | 20.34 | 10.47 | 51.15 |
| Pre-QKV-Post | 2.74 | 20.13 | 10.37 | 52.65 |
| Pre-QKV-Pre | 2.74 | 19.97 | 10.33 | 53.05 |
| Pre-QK-Pre | 2.75 | 20.22 | 10.43 | 52.29 |
| QKV-Pre | 2.74 | 19.96 | 10.33 | 52.57 |

| Methods | Loss↓ | C4↓ | Pile↓ | HS↑ |
|---------|-------|-----|-------|-----|
| Post-Norm | 2.76 | 20.43 | 10.57 | 51.20 |
| Pre-Norm | 2.75 | 20.30 | 10.48 | 51.97 |
| QK-norm | 2.75 | 20.22 | 10.43 | 52.29 |
| Mix-LN | 2.76 | 20.43 | 10.56 | 51.29 |
| Post-Pre | 2.75 | 20.26 | 10.46 | 51.19 |
| Pre-Post | 2.74 | 20.15 | 10.40 | 52.42 |
| HybridNorm | 2.74 | 20.00 | 10.29 | 53.35 |
| HybridNorm* | **2.73** | **19.85** | **10.25** | **53.36** |

As shown in Table 6, HybridNorm (a.k.a. QKV-Post) and its variant HybridNorm* consistently surpass other methods. Notably, HybridNorm* achieves the lowest training loss and perplexity while attaining the highest accuracy on HellaSwag. Specifically, by comparing HybridNorm with the left first block in Table 6, we find that QKV-Norm is the most effective normalization. Similarly, comparing HybridNorm with the left second block, we observe that combining QKV-Norm with Post-Norm in the FFN yields superior performance. From the right table, one can see that the Pre-Post configuration indeed leads to improved performance, while replacing Pre-Norm in the MHA with QKV-Norm to form HybridNorm further enhances performance, achieving the best results.

**Special Treatment of First Block** For the special treatment of the first block, we test different architectures, such as adding a normalization layer after embedding (call EmbedNorm) and armed the first block with QKV-norm and Pre-Norm in FFN (call First-QKV-Pre), which formulations are:

$$Y^0 = \text{MHA}_{QKV}(X^0) + X^0, \quad X^1 = \text{FFN}(\text{Norm}(Y^0)) + Y^0. \tag{10}$$

As shown in Figure 5, we can see that, except for EmbedNorm, the special treatment of the first block effectively reduces training loss and improves downstream performance.

## 5.5 Scaling Laws Experiments

We compare the loss scaling curves between Pre-Norm and HybridNorm* across a range of dense model sizes, from 151M to 1.2B parameters. The model sizes used for the scaling law experiments are detailed in Table 9, and all models are trained using the same setting and hyperparameters for fair comparison, as specified in Table 10. Models with 151M, 285M, 550M, and 1.2B parameters are trained on 200B, 200B, 300B, and 1T tokens, respectively. As shown in Figure 6, HybridNorm* exhibits superior scaling properties, demonstrating lower training loss as the model size increases. This highlights its capacity to maintain both training stability and performance, even for extremely large models, thereby making it highly suitable for scaling to billion-parameter regimes.

## 5.6 Deeper Models

To further evaluate the robustness of HybridNorm and HybridNorm* in deeper architectures, we conduct experiments on transformers with depths ranging from 16 to 29 layers while maintaining

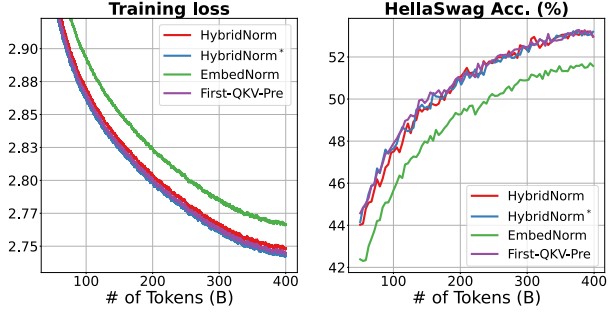
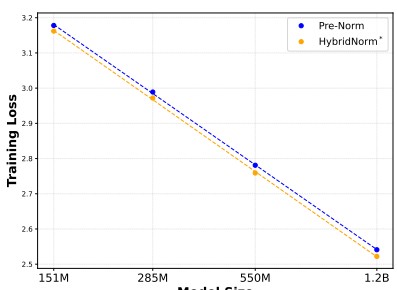

Figure 5: Training loss and accuracy on HellaSwag of 550M dense models with different normalization methods for the first block.

Figure 6: Scaling law curves of Pre-Norm and HybridNorm*.

Table 7: Performance of dense models with different depths under 400B training tokens. We report the training loss, the perplexity on C4 and Pile, and the accuracy on HellaSwag and PIQA.

| Methods | 550M, 16 Layers | | | | | 543M, 29 Layers | | | | |
| | Loss↓ | C4↓ | Pile↓ | HellaSwag↑ | PIQA↑ | Loss↓ | C4↓ | Pile↓ | HellaSwag↑ | PIQA↑ |
| --- | --- | --- | --- | --- | --- | --- | --- | --- | --- | --- |
| Post-Norm | 2.76 | 20.43 | 10.57 | 51.20 | 71.80 | diverge | | | | |
| Pre-Norm | 2.75 | 20.30 | 10.48 | 51.97 | 71.14 | 2.73 | 19.88 | 10.31 | 53.86 | 71.63 |
| HybridNorm | 2.74 | 20.00 | 10.29 | 53.35 | **71.96** | 2.72 | 19.67 | 10.18 | **54.89** | **72.62** |
| HybridNorm* | **2.73** | **19.85** | **10.25** | **53.36** | 71.15 | **2.71** | **19.52** | **10.10** | 54.54 | 72.01 |

a comparable parameter budget. This setup allows for a fair comparison of different normalization strategies in deep transformer architectures. As shown in Table 7, both HybridNorm and HybridNorm* consistently outperform Pre-Norm and Post-Norm across various depths, demonstrating their effectiveness in stabilizing deep model training. A particularly striking observation is that Post-Norm fails to converge at 29 layers, reinforcing its well-documented instability in deeper architectures. In contrast, HybridNorm and HybridNorm* not only ensure stable training across all depths but also achieve significantly lower training loss and perplexity on both the C4 and Pile datasets. These improvements indicate that HybridNorm-based normalization strategies mitigate optimization difficulties that commonly arise in deep transformers. Furthermore, HybridNorm* achieves the highest accuracy on challenging downstream benchmarks such as HellaSwag and PIQA, suggesting that its benefits extend beyond mere training stability to enhanced generalization on real-world tasks. These results provide strong empirical evidence that HybridNorm-based normalization schemes enable deeper transformer training while preserving superior optimization efficiency and downstream task performance.

## 6 Conclusion

In this paper, we have introduced HybridNorm, a novel hybrid normalization strategy that has seamlessly integrated the advantages of both Pre-Norm and Post-Norm, thereby addressing the longstanding trade-offs in transformer training. We have provided both comprehensive theoretical and empirical analyses to demonstrate how HybridNorm has stabilized gradient propagation while preserving strong regularization effects, ultimately improving both convergence speed and final model performance. Extensive experiments across diverse benchmarks have substantiated the effectiveness of our approach, consistently showing that HybridNorm has outperformed conventional normalization schemes in terms of stability and accuracy. These findings have highlighted the importance of re-examining the role and placement of normalization within transformer architectures, paving the way for further exploration of hybrid normalization paradigms. We believe that HybridNorm has marked a significant step forward in the development of more robust and efficient transformer models, offering practical advantages for training next-generation large-scale neural networks.

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

# Appendix

## Table of Contents

# A Related Work

**Architecture Modifications in Transformers**   Recent efforts in transformer architecture modifications have sought to optimize both the computational efficiency and the expressiveness of the model. These efforts include changes to the attention mechanism and feed-forward networks all aimed at improving performance on a variety of tasks, ranging from language modeling to vision tasks [35, 3]. For example, Multi-head Latent Attention (MLA) [12], Mixture of Experts (MoE) [36]. While these modifications contribute to more efficient training, they also require careful integration with other components, such as normalization layers, to maintain model stability and performance.

**Normalization Types in Transformers**   Normalization layers are integral to the success of deep learning models, and transformers are no exception. The most commonly used normalization technique in transformers is LayerNorm [5], which normalizes the activations of each layer independently. However, alternative methods such as RMSNorm [8], which normalizes using root mean square statistics, have been proposed as more effective alternatives in certain settings. These methods are designed to mitigate the challenges of internal covariate shift and gradient instability, which are critical for the success of large-scale transformer models.

**Normalization Settings in Attention**   For training stability, QK-Norm [37, 38] modifies the standard attention mechanism by applying normalization directly to the query (Q) and key (K) components during attention computation. Building upon this, QKV-Norm [9, 10] extends the approach by normalizing the Query (Q), Key (K), and Value (V) components. This comprehensive normalization ensures that all critical components of the attention mechanism are normalized, resulting in enhanced stability and improved performance.

**Location of Normalization Layers**   Recent research has also explored the impact of normalization location in both Vision Transformers [39, 40] and language models [41, 40]. For example, the choice between Pre-Norm and Post-Norm architectures has been widely studied in the transformer literature [35, 42, 41]. Pre-Norm, where normalization is applied before the residual connection, has been shown to be more stable in deep networks and accelerates convergence [4]. Although Post-Norm is more challenging to train, it tends to deliver better final performance by normalizing after the residual connection [40]. DeepNorm [7] was proposed as a strategy to address training instability in deep transformers, which scales the residual connections by a carefully chosen factor to improve gradient flow and mitigate exploding or vanishing gradients. Ding et al. [43] introduced Sandwich-LN in multimodal settings to improve training stability, a strategy that has also been adopted by the Gemma team in their recent models [44]. Similarly, OLMo-2 [30] applies the normalization layer after the sublayer but before the residual connection, differing from both traditional Pre-LN and Post-LN schemes. The method most similar to ours is Mix-LN [11], which applies Post-Norm to the earlier layers and Pre-Norm to the deeper layers, achieving improved training stability and better performance. In contrast, our HybridNorm integrates Pre-Norm and Post-Norm within each transformer block. This intra-layer hybridization offers several key advantages: (1) consistently improved model performance, (2) intra-layer hybridization ensures uniformity across all layers, facilitating other post-training such as pruning and quantization.

# B Theoretical Gradient Analysis

For simplicity, we consider a single-head attention layer. The input is $X \in \mathbb{R}^{s \times d}$, representing a sequence of $s$ tokens with dimension $d$. Throughout this section, we denote RMSNorm[1] as $\mathrm{Norm}(\cdot)$, i.e., $\mathrm{Norm}(\boldsymbol{x}) = \alpha \odot \frac{\boldsymbol{x}}{\mathrm{RMS}(\boldsymbol{x})}$ for $\boldsymbol{x} \in \mathbb{R}^d$, where $\mathrm{RMS}(\boldsymbol{x}) = \sqrt{(x_1^2 + \cdots + x_d^2)/d}$. For further simplicity, we set $\alpha = \mathbf{1}_d$. The learnable parameters $W_Q, W_K, W_V \in \mathbb{R}^{d \times d_k}$ and $W_O \in \mathbb{R}^{d_k \times d}$. Let $X_N = \mathrm{Norm}(X)$, $M = \frac{1}{\sqrt{d_k}} X_N W_Q W_K^\top X_N^\top$, and $A = \mathrm{softmax}(M)$. The output of the attention block with Pre-Norm is then given by

$$S = A X_N W_V W_O. \tag{11}$$

---

[1]Given that the vast majority of popular LLMs are based on RMSNorm, our experiments and conclusions are broadly applicable to standard LLM architectures. Futher, in Appendix B.5, we demonstrate that RMSNorm and LayerNorm exhibit no fundamental differences, both in theoretical analysis and empirical observations.

Defining $Q = XW_Q$, $K = XW_K$, and $V = XW_V$, with their normalized counterparts $Q_N = \text{Norm}(Q)$, $K_N = \text{Norm}(K)$, and $V_N = \text{Norm}(V)$, the output of the attention block with QKV-Norm is formulated as

$$S_N = A_N V_N W_O, \tag{12}$$

where $A_N = \text{softmax}(M_N)$ and $M_N = \frac{1}{\sqrt{d_k}} Q_N K_N^\top$.

Defining $\hat{Q} = X_N W_Q$ and $\hat{K}_N = X_N W_K$, with their normalized counterparts $\hat{Q}_N = \text{Norm}(\hat{Q})$ and $\hat{K}_N = \text{Norm}(\hat{K})$, the output of the attention block with Pre-Norm and QK-Norm is formulated as

$$\hat{S} = \hat{A}_N X_N W_V W_O, \tag{13}$$

where $\hat{A}_N = \text{softmax}(\hat{M}_N)$ and $\hat{M}_N = \frac{1}{\sqrt{d_k}} \hat{Q}_N \hat{K}_N^\top$.

Following the prior work of Noci et al. [45], Ormaniec et al. [46], we analyze the gradients by computing derivatives using row-wise vectorization and arranging the Jacobian in the numerator layout, i.e.,

$$\frac{\partial Y}{\partial X} = \frac{\partial \text{vec}_r(Y)}{\partial \text{vec}_r(X)^\top}.$$

The following derivation primarily relies on the chain rule and the following rule

$$\frac{\partial AWB}{\partial W} = A \otimes B^\top, \tag{14}$$

where $A \in \mathbb{R}^{m \times n}, W \in \mathbb{R}^{n \times p}, B \in \mathbb{R}^{p \times q}$, and $\otimes$ is the Kronecker product. The proof of Eq. 14 can be found in [47].

## B.1 Theorem 2

We first present the following extension of Lemma 2 in Noci et al. [45].

**Lemma 1** (Extention of Lemma 2 in Noci et al. [45])**.** *The gradients of the attention with Pre-Norm defined in Eq. (11) are given by*

$$\frac{\partial S}{\partial W_O} = \text{softmax}\left(\frac{X_N W_Q W_K^\top X_N^\top}{\sqrt{d_k}}\right) X_N W_V \otimes I_d, \tag{15}$$

$$\frac{\partial S}{\partial W_V} = \text{softmax}\left(\frac{X_N W_Q W_K^\top X_N^\top}{\sqrt{d_k}}\right) X_N \otimes W_O^\top, \tag{16}$$

$$\frac{\partial S}{\partial W_Q} = \left(I_s \otimes W_O^\top W_V^\top X_N^\top\right) \frac{\partial A}{\partial M} \left(\frac{X_N \otimes X_N W_K}{\sqrt{d_k}}\right), \tag{17}$$

$$\frac{\partial S}{\partial W_K} = \left(I_s \otimes W_O^\top W_V^\top X_N^\top\right) \frac{\partial A}{\partial M} \left(\frac{X_N W_Q \otimes X_N}{\sqrt{d_k}}\right) K_{d_k,d}, \tag{18}$$

*where the gradients of the softmax with respect to its inputs is*

$$\frac{\partial A}{\partial M} = \text{blockdiag}\left(\frac{\partial A_{i,:}}{\partial M_{i,:}}\right) = \text{blockdiag}\left(\text{diag}(A_{i,:}) - A_{i,:} A_{i,:}^\top\right), \tag{19}$$

$A_{i,:}$ *is the $i$-th row of $A$ in column vector format, and the commutation matrix $K_{d_k,d}$ is a permutation matrix that transforms the row-wise vectorization of $W_K$ into the column-wise vectorization of $W_K$, i.e.,*

$$K_{d_k,d} \text{vec}_r(W_K) = \text{vec}_r(W_K^\top).$$

The gradient of $X_N$ with respect to $X$ is

$$\frac{\partial X_N}{\partial X} = \frac{\partial \text{Norm}(X)}{\partial X} = \text{blockdiag}\left(\frac{\partial \text{Norm}(X_{i,:})}{\partial X_{i,:}}\right) = \text{blockdiag}\left(\frac{\sqrt{d}}{\|X_{i,:}\|_2}\left(I_d - \frac{X_{i,:} X_{i,:}^\top}{\|X_{i,:}\|_2^2}\right)\right), \tag{20}$$

where $X_{i,:}$ is the $i$-th row of $X$ represented as a column vector. The definitions of $\frac{\partial Q_N}{\partial Q}$, $\frac{\partial K_N}{\partial K}$, $\frac{\partial V_N}{\partial V}$, $\frac{\partial \hat{Q}_N}{\partial \hat{Q}}$, and $\frac{\partial \hat{K}_N}{\partial \hat{K}}$ follow similarly, *i.e.*, for $\bullet \in \{Q, K, V, \hat{Q}, \hat{K}\}$,

$$\frac{\partial \bullet_N}{\partial \bullet} = \text{blockdiag} \left( \frac{\sqrt{d_k}}{\|\bullet_{i,:}\|_2} \left( I_{d_k} - \frac{\bullet_{i,:}\bullet_{i,:}^\top}{\|\bullet_{i,:}\|_2^2} \right) \right). \tag{21}$$

**Lemma 2.** *The gradients of the attention with QKV-Norm defined in Eq. (12) are*

$$\frac{\partial S_N}{\partial W_O} = \text{softmax} \left( \frac{\text{Norm}(XW_Q)\text{Norm}(XW_K)^\top}{\sqrt{d_k}} \right) \text{Norm}(XW_V) \otimes I_d, \tag{22}$$

$$\frac{\partial S_N}{\partial W_V} = \left( \text{softmax} \left( \frac{\text{Norm}(XW_Q)\text{Norm}(XW_K)^\top}{\sqrt{d_k}} \right) \otimes W_O^\top \right) \frac{\partial V_N}{\partial V} (X \otimes I_{d_k}), \tag{23}$$

$$\frac{\partial S_N}{\partial W_Q} = \left( I_s \otimes W_O^\top \text{Norm}(XW_V)^\top \right) \frac{\partial A_N}{\partial M_N} \left( \frac{I_s \otimes \text{Norm}(XW_K)}{\sqrt{d_k}} \right) \frac{\partial Q_N}{\partial Q} (X \otimes I_{d_k}), \tag{24}$$

$$\frac{\partial S_N}{\partial W_K} = \left( I_s \otimes W_O^\top \text{Norm}(XW_V)^\top \right) \frac{\partial A_N}{\partial M_N} \left( \frac{\text{Norm}(XW_Q) \otimes I_s}{\sqrt{d_k}} \right) K_{d_k,s} \frac{\partial K_N}{\partial K} (X \otimes I_{d_k}), \tag{25}$$

*where the definition of $\frac{\partial A_N}{\partial M_N}$ is similar to $\frac{\partial A}{\partial M}$ and $K_{d_k,s}$ is the commutation matrix s.t., $K_{d_k,s}\text{vec}_r(K_N) = \text{vec}_r(K_N^\top)$.*

The proof of Lemma 2 is provided in Appendix B.2. Similarly, for the attention with Pre-Norm and QK-Norm, we derive the following lemma.

**Lemma 3.** *The gradients of the attention with Pre-Norm and QK-Norm defined in Eq. (13) are*

$$\frac{\partial \hat{S}}{\partial W_O} = \text{softmax} \left( \frac{\text{Norm}(X_N W_Q)\text{Norm}(X_N W_K)^\top}{\sqrt{d_k}} \right) X_N W_V \otimes I_d, \tag{26}$$

$$\frac{\partial \hat{S}}{\partial W_V} = \text{softmax} \left( \frac{\text{Norm}(X_N W_Q)\text{Norm}(X_N W_K)^\top}{\sqrt{d_k}} \right) X_N \otimes W_O^\top, \tag{27}$$

$$\frac{\partial \hat{S}}{\partial W_Q} = \left( I_s \otimes W_O^\top W_V^\top X_N^\top \right) \frac{\partial \hat{A}_N}{\partial \hat{M}_N} \left( \frac{I_s \otimes \text{Norm}(X_N W_K)}{d_k} \right) \frac{\partial \hat{Q}_N}{\partial \hat{Q}} (X_N \otimes I_{d_k}), \tag{28}$$

$$\frac{\partial \hat{S}}{\partial W_K} = \left( I_s \otimes W_O^\top W_V^\top X_N^\top \right) \frac{\partial \hat{A}_N}{\partial \hat{M}_N} \left( \frac{\text{Norm}(X_N W_Q) \otimes I_s}{\sqrt{d_k}} \right) \hat{K}_{d_k,s} \frac{\partial \hat{K}_N}{\partial \hat{K}} (X_N \otimes I_{d_k}), \tag{29}$$

*where the definition of $\frac{\partial \hat{A}_N}{\partial \hat{M}_N}$ is similar to $\frac{\partial A}{\partial M}$ and $\hat{K}_{d_k,s}$ is the commutation matrix s.t., $\hat{K}_{d_k,s}\text{vec}_r(\hat{K}_N) = \text{vec}_r(\hat{K}_N^\top)$.*

The proof of Lemma 3 can be found in Appendix B.3.

Armed with the above lemmas, we arrive at the following theorem, which characterizes the gradient norms of Pre-Norm, Pre-Norm with QK-Norm, and QKV-Norm.

**Theorem 2.** *For the attention with Pre-Norm, we have*

$$\left\| \frac{\partial S}{\partial W_O} \right\|_F = \mathcal{O} \left( s\sqrt{d}\|W_V\|_2 \right), \tag{30}$$

$$\left\| \frac{\partial S}{\partial W_V} \right\|_F = \mathcal{O} \left( s\|W_O\|_F \right), \tag{31}$$

$$\left\| \frac{\partial S}{\partial W_Q} \right\|_F = \mathcal{O} \left( \frac{(s)^{3/2}}{2\sqrt{d_k}} \|W_K\|_2\|W_V\|_2\|W_O\|_2 \right), \tag{32}$$

$$\left\| \frac{\partial S}{\partial W_K} \right\|_F = \mathcal{O} \left( \frac{(s)^{3/2}}{\sqrt{d_k}} \|W_Q\|_2\|W_V\|_2\|W_O\|_2 \right). \tag{33}$$

*For the attention with Pre-Norm and QK-Norm, we have*

$$\left\| \frac{\partial \hat{S}}{\partial W_O} \right\|_F = \mathcal{O} \left( s\sqrt{d}\|W_V\|_2 \right), \tag{34}$$

$$\left\| \frac{\partial \hat{S}}{\partial W_V} \right\|_F = \mathcal{O}\left( s \|W_O\|_F \right), \tag{35}$$

$$\left\| \frac{\partial \hat{S}}{\partial W_Q} \right\|_F = \mathcal{O}\left( \frac{s\sqrt{sd_k}}{\sigma_{\min}^Q} \|W_V\|_2 \|W_O\|_2 \right), \tag{36}$$

$$\left\| \frac{\partial \hat{S}}{\partial W_K} \right\|_F = \mathcal{O}\left( \frac{s\sqrt{sd_k}}{\sigma_{\min}^K} \|W_V\|_2 \|W_O\|_2 \right). \tag{37}$$

*For the attention with QKV-Norm, we have*

$$\left\| \frac{\partial S_N}{\partial W_O} \right\|_F = \mathcal{O}\left( s\sqrt{d} \right), \tag{38}$$

$$\left\| \frac{\partial S_N}{\partial W_V} \right\|_F = \mathcal{O}\left( \frac{sd_k}{\sigma_{\min}^V} \|W_O\|_2 \right), \tag{39}$$

$$\left\| \frac{\partial S_N}{\partial W_Q} \right\|_F = \mathcal{O}\left( \frac{s\sqrt{sd_k}}{\sigma_{\min}^Q} \|W_O\|_2 \right), \tag{40}$$

$$\left\| \frac{\partial S_N}{\partial W_K} \right\|_F = \mathcal{O}\left( \frac{s\sqrt{sd_k}}{\sigma_{\min}^K} \|W_O\|_2 \right), \tag{41}$$

*where $\sigma_{\min}^Q, \sigma_{\min}^K, \sigma_{\min}^V$ are minimal singular value of $W_Q, W_K, W_V$, respectively.*

Theorem 2 presents the gradient norms of various methods, and its proof is provided in Appendix B.4. In the attention with Pre-Norm, the gradient of the weight matrix exhibits strong dependencies on other weights; for instance, $W_Q$ and $W_K$ are influenced by all three other weight matrices but not by themselves. In contrast, in the attention with QKV-Norm, the gradient of each weight matrix depends at most on itself and $W_O$. This suggests that the gradient of the attention with Pre-Norm is more tightly coupled with other weight matrices compared to the gradient of the attention with QKV-Norm. Whereas the attention with Pre-Norm and QK-Norm lies between the two methods. Therefore, during the gradient optimization process, if the norm of a certain weight becomes excessively large, it is more challenging to control in the attention with Pre-Norm, leading to an increase in gradient magnitude. This, in turn, creates a vicious cycle that may result in model collapse. In contrast, the attention with QKV-Norm alleviates this issue to some extent, which significantly benefits the stability of model training. Regarding the degree of coupling, the relationship follows Pre-Norm > Pre-Norm with QK-Norm > QKV-Norm, whereas for training stability, the hierarchy is reversed: Pre-Norm < Pre-Norm with QK-Norm < QKV-Norm.

## B.2 Proof of Lemma 2

*Proof of Lemma 2.* For $\frac{\partial S_N}{\partial W_O}$, according to Eq. (14), we obtain

$$\frac{\partial S_N}{\partial W_O} = A_N V_N \otimes I_d = \text{softmax}\left( \frac{\text{Norm}(XW_Q)\text{Norm}(XW_K)^\top}{\sqrt{d_k}} \right) \text{Norm}(XW_V) \otimes I_d.$$

For $\frac{\partial S_N}{\partial W_V}$, using the chain rule and Eq. (14), we have

$$\begin{aligned}
\frac{\partial S_N}{\partial W_V} &= \frac{\partial S_N}{\partial V_N} \frac{\partial V_N}{\partial V} \frac{\partial V}{\partial W_V} \\
&= (A \otimes W_O^\top) \frac{\partial V_N}{\partial V} (X \otimes I_{d_k}) \\
&= \left( \text{softmax}\left( \frac{\text{Norm}(XW_Q)\text{Norm}(XW_K)^\top}{\sqrt{d_k}} \right) \otimes W_O^\top \right) \frac{\partial V_N}{\partial V} (X \otimes I_{d_k}).
\end{aligned}$$

For $\frac{\partial S_N}{\partial W_Q}$, using the chain rule and Eq. (14), we obtain

$$\frac{\partial S_N}{\partial W_Q} = \frac{\partial S_N}{\partial A_N} \frac{\partial A_N}{\partial M_N} \frac{\partial M_N}{\partial Q_N} \frac{\partial Q_N}{\partial Q} \frac{\partial Q}{W_Q}$$

$$= \left(I_s \otimes W_O^\top V_N^\top\right) \frac{\partial A_N}{\partial M_N} \left(\frac{I_s \otimes K_N}{\sqrt{d_k}}\right) \frac{\partial Q_N}{\partial Q}(X \otimes I_{d_k})$$

$$= \left(I_s \otimes W_O^\top \text{Norm}(XW_V)^\top\right) \frac{\partial A_N}{\partial M_N} \left(\frac{I_s \otimes \text{Norm}(XW_K)}{d_k}\right) \frac{\partial Q_N}{\partial Q}(X \otimes I_{d_k}).$$

Similarly, for $\frac{\partial S_N}{\partial W_K}$, we have

$$\frac{\partial S_N}{\partial W_K} = \frac{\partial S_N}{\partial A_N} \frac{\partial A_N}{\partial M_N} \frac{\partial M_N}{\partial K_N} \frac{\partial K_N}{\partial K} \frac{\partial K}{W_K}$$

$$= \left(I_s \otimes W_O^\top V_N^\top\right) \frac{\partial A_N}{\partial M_N} \left(\frac{Q_N \otimes I_s}{d_k}\right) \frac{\partial \text{vec}_r(K_N^\top)}{\partial \text{vec}_r(K_N)^\top} \frac{\partial K_N}{\partial K}(X \otimes I_{d_k})$$

$$= \left(I_s \otimes W_O^\top \text{Norm}(XW_V)^\top\right) \frac{\partial A_N}{\partial M_N} \left(\frac{\text{Norm}(XW_Q) \otimes I_s}{\sqrt{d_k}}\right) K_{d_k,s} \frac{\partial K_N}{\partial K}(X \otimes I_{d_k}).$$

$\square$

## B.3 Proof of Lemma 3

*Proof of Lemma 3.* For $\frac{\partial \hat{S}}{\partial W_O}$, according to Eq. (14), we obtain

$$\frac{\partial \hat{S}}{\partial W_O} = \hat{A}_N X_N W_V \otimes I_d = \text{softmax}\left(\frac{\text{Norm}(X_N W_Q)\text{Norm}(X_N W_K)^\top}{\sqrt{d_k}}\right) X_N W_V \otimes I_d.$$

For $\frac{\partial \hat{S}}{\partial W_V}$, using Eq. (14), we have

$$\frac{\partial \hat{S}}{\partial W_V} = \hat{A}_N X_N \otimes W_O^\top = \text{softmax}\left(\frac{\text{Norm}(X_N W_Q)\text{Norm}(X_N W_K)^\top}{\sqrt{d_k}}\right) X_N \otimes W_O^\top.$$

For $\frac{\partial \hat{S}}{\partial W_Q}$, using the chain rule and Eq. (14), we obtain

$$\frac{\partial \hat{S}}{\partial W_Q} = \frac{\partial \hat{S}}{\partial \hat{A}_N} \frac{\partial \hat{A}_N}{\partial \hat{M}_N} \frac{\partial \hat{M}_N}{\partial \hat{Q}_N} \frac{\partial \hat{Q}_N}{\partial \hat{Q}} \frac{\partial \hat{Q}}{W_Q}$$

$$= \left(I_s \otimes W_O^\top W_V^\top X_N^\top\right) \frac{\partial \hat{A}_N}{\partial \hat{M}_N} \left(\frac{I_s \otimes \hat{K}_N}{\sqrt{d_k}}\right) \frac{\partial \hat{Q}_N}{\partial \hat{Q}}(X_N \otimes I_{d_k})$$

$$= \left(I_s \otimes W_O^\top W_V^\top X_N^\top\right) \frac{\partial \hat{A}_N}{\partial \hat{M}_N} \left(\frac{I_s \otimes \text{Norm}(X_N W_K)}{d_k}\right) \frac{\partial \hat{Q}_N}{\partial \hat{Q}}(X_N \otimes I_{d_k}).$$

Similarly, for $\frac{\partial \hat{S}}{\partial W_K}$, we have

$$\frac{\partial \hat{S}}{\partial W_K} = \frac{\partial \hat{S}}{\partial \hat{A}_N} \frac{\partial \hat{A}_N}{\partial \hat{M}_N} \frac{\partial \hat{M}_N}{\partial \hat{K}_N} \frac{\partial \hat{K}_N}{\partial \hat{K}} \frac{\partial \hat{K}}{W_K}$$

$$= \left(I_s \otimes W_O^\top W_V^\top X_N^\top\right) \frac{\partial \hat{A}_N}{\partial \hat{M}_N} \left(\frac{\hat{Q}_N \otimes I_s}{d_k}\right) \frac{\partial \text{vec}_r(\hat{K}_N^\top)}{\partial \text{vec}_r(\hat{K}_N)^\top} \frac{\partial \hat{K}_N}{\partial \hat{K}}(X_N \otimes I_{d_k})$$

$$= \left(I_s \otimes W_O^\top W_V^\top X_N^\top\right) \frac{\partial \hat{A}_N}{\partial \hat{M}_N} \left(\frac{\text{Norm}(X_N W_Q) \otimes I_s}{\sqrt{d_k}}\right) K_{d_k,s} \frac{\partial \hat{K}_N}{\partial \hat{K}}(X_N \otimes I_{d_k}).$$

$\square$

## B.4 Proof of Theorem 2

The proof is primarily based on the following facts

- $\text{tr}(B \otimes C) = \text{tr}(B)\text{tr}(C)$

- $(B \otimes C)(D \otimes E) = (BD) \otimes (CE)$
- $\|B \otimes C\|_F = \|B\|_F \|C\|_F$
- $\|B \otimes C\|_2 = \|B\|_2 \|C\|_2$
- $\|BC\|_2 \leq \|B\|_2 \|C\|_2$
- $\|BC\|_F \leq \|B\|_2 \|C\|_F \leq \|B\|_F \|C\|_F$
- $\|X_N\|_F = \sqrt{s}$
- If $\boldsymbol{p} \in \mathbb{R}^s$, $p_i \geq 0$ and $\sum_{i=1}^{s} p_i = 1$, then $\|\mathrm{diag}(\boldsymbol{p}) - \boldsymbol{p}\boldsymbol{p}^\top\|_2 \leq \frac{1}{2}$.

  *Proof.* According to Gershgorin Circle Theorem [48], every eigenvalue of $\mathrm{diag}(\boldsymbol{p}) - \boldsymbol{p}\boldsymbol{p}^\top$ lies within

  $$
  \bigcup_{i=1}^{s} [p_i - p_i^2 - \sum_{j \neq i} p_i p_j, p_i - p_i^2 + \sum_{j \neq i} p_i p_j]
  $$

  $$
  = \bigcup_{i=1}^{s} [p_i(1 - p_i) - p_i \sum_{j \neq i} p_j, p_i(1 - p_i) + p_i \sum_{j \neq i} p_j]
  $$

  $$
  = \bigcup_{i=1}^{s} [p_i(1 - p_i) - p_i(1 - p_i), p_i(1 - p_i) + p_i(1 - p_i)]
  $$

  $$
  = \bigcup_{i=1}^{s} [0, 2p_i(1 - p_i)]
  $$

  $$
  \subseteq [0, \frac{1}{2}].
  $$

  Therefore, $\|\mathrm{diag}(\boldsymbol{p}) - \boldsymbol{p}\boldsymbol{p}^\top\|_2 \leq \frac{1}{2}$. When $p_1 = p_2 = \frac{1}{2}$, the equality holds, indicating that this bound is tight. $\qquad\square$

- If $A \in \mathbb{R}^{s \times s}$ is a stochastic matrix, *i.e.*, $A\mathbf{1}_s = \mathbf{1}_s$ and each entry is nonnegative, then $\|A\|_2 \leq \|A\|_F \leq \sqrt{s}$.

  *Proof.* Note that

  $$
  \|A\|_F = \sqrt{\sum_{i=1}^{s}\sum_{j=1}^{s} a_{ij}^2} \leq \sqrt{\sum_{i=1}^{s}\sum_{j=1}^{s} a_{ij}} = \sqrt{\sum_{i=1}^{s} 1} = \sqrt{s}.
  $$

  If $A = \mathbf{1}_s e_1$, then

  $$
  \|A\|_2 = \sqrt{\lambda_{\max}(A^\top A)} = \sqrt{\lambda_{\max}(s e_1 e_1^\top)} = \sqrt{s}.
  $$

  Hence, the bound is tight. $\qquad\square$

*Proof of Theorem 2.* According to fundamental algebraic operations and Lemma 1, we obtain

$$
\left\|\frac{\partial S}{\partial W_O}\right\|_F = \|AX_N W_V \otimes I_d\|_F = \|AX_N W_V\|_F \|I_d\|_F \leq \sqrt{d}\|A\|_2 \|X_N\|_F \|W_V\|_2 \leq s\sqrt{d}\|W_V\|_2,
$$

$$
\left\|\frac{\partial S}{\partial W_V}\right\|_F = \|AX_N \otimes W_O^\top\|_F = \|AX_N\|_F \|W_O\|_F \leq \|A\|_2 \|X_N\|_F \|W_O\|_F \leq s\|W_O\|_F,
$$

$$
\left\|\frac{\partial S}{\partial W_Q}\right\|_F = \left\|(I_s \otimes W_O^\top W_V^\top X_N^\top)\frac{\partial A}{\partial M}\left(\frac{X_N \otimes X_N W_K}{\sqrt{d_k}}\right)\right\|_F
$$

$$
= \frac{1}{\sqrt{d_k}}\left\|(I_s \otimes W_O^\top W_V^\top X_N^\top)\,\mathrm{blockdiag}(\mathrm{diag}(A_{i,:}) - A_{i,:}A_{i,:}^\top)(X_N \otimes X_N W_K)\right\|_F
$$

$$= \frac{1}{\sqrt{d_k}} \left\| \text{blockdiag}((W_O^\top W_V^\top X_N^\top)(\text{diag}(A_{i,:}) - A_{i,:}A_{i,:}^\top))(X_N \otimes X_N W_K) \right\|_F$$

$$= \frac{1}{\sqrt{d_k}} \left\| \begin{bmatrix} X_{N_{1,:}}^\top \otimes W_O^\top W_V^\top X_N^\top(\text{diag}(A_{1,:}) - A_{1,:}A_{1,:}^\top))X_N W_K \\ \vdots \\ X_{N_{s,:}}^\top \otimes W_O^\top W_V^\top X_N^\top(\text{diag}(A_{s,:}) - A_{s,:}A_{s,:}^\top))X_N W_K \end{bmatrix} \right\|_F$$

$$\leq \frac{1}{\sqrt{d_k}} \|X_N\|_F \|W_O\|_2 \|W_V\|_2 \|X_N\|_2 \frac{1}{2} \|W_K\|_2 \|X_N\|_F$$

$$\leq \frac{1}{2\sqrt{d_k}} \|W_K\|_2 \|W_V\|_2 \|W_O\|_2 \|X_N\|_F^3$$

$$= \frac{(s)^{3/2}}{2\sqrt{d_k}} \|W_K\|_2 \|W_V\|_2 \|W_O\|_2,$$

$$\left\| \frac{\partial S}{\partial W_K} \right\|_F = \left\| (I_s \otimes W_O^\top W_V^\top X_N^\top) \frac{\partial A}{\partial M} \left( \frac{X_N W_Q \otimes X_N}{\sqrt{d_k}} \right) K_{d_k,d} \right\|_F$$

$$= \left\| (I_s \otimes W_O^\top W_V^\top X_N^\top) \frac{\partial A}{\partial M} \left( \frac{X_N W_Q \otimes X_N}{\sqrt{d_k}} \right) \right\|_F$$

$$= \frac{1}{\sqrt{d_k}} \left\| (I_s \otimes W_O^\top W_V^\top X_N^\top) \text{blockdiag}(\text{diag}(A_{i,:}) - A_{i,:}A_{i,:}^\top)(X_N W_Q \otimes X_N) \right\|_F$$

$$= \frac{1}{\sqrt{d_k}} \left\| \text{blockdiag}((W_O^\top W_V^\top X_N^\top)(\text{diag}(A_{i,:}) - A_{i,:}A_{i,:}^\top))(X_N W_Q \otimes X_N) \right\|_F$$

$$= \frac{1}{\sqrt{d_k}} \left\| \begin{bmatrix} (X_N W_Q)_{1,:}^\top \otimes W_O^\top W_V^\top X_N^\top(\text{diag}(A_{1,:}) - A_{1,:}A_{1,:}^\top))X_N \\ \vdots \\ (X_N W_Q)_{s,:}^\top \otimes W_O^\top W_V^\top X_N^\top(\text{diag}(A_{s,:}) - A_{s,:}A_{s,:}^\top))X_N \end{bmatrix} \right\|_F$$

$$\leq \frac{1}{\sqrt{d_k}} \|W_Q\|_2 \|X_N\|_F \|W_O\|_2 \|W_V\|_2 \|X_N\|_2 \frac{1}{2} \|X_N\|_F$$

$$\leq \frac{1}{2\sqrt{d_k}} \|W_Q\|_2 \|W_V\|_2 \|W_O\|_2 \|X_N\|_F^3$$

$$= \frac{(s)^{3/2}}{2\sqrt{d_k}} \|W_Q\|_2 \|W_V\|_2 \|W_O\|_2.$$

Therefore,

$$\left\| \frac{\partial S}{\partial W_O} \right\|_F = \mathcal{O}\left( s\sqrt{d} \|W_V\|_2 \right),$$

$$\left\| \frac{\partial S}{\partial W_V} \right\|_F = \mathcal{O}\left( s \|W_O\|_F \right),$$

$$\left\| \frac{\partial S}{\partial W_Q} \right\|_F = \mathcal{O}\left( \frac{(s)^{3/2}}{2\sqrt{d_k}} \|W_K\|_2 \|W_V\|_2 \|W_O\|_2 \right),$$

$$\left\| \frac{\partial S}{\partial W_K} \right\|_F = \mathcal{O}\left( \frac{(s)^{3/2}}{\sqrt{d_k}} \|W_Q\|_2 \|W_V\|_2 \|W_O\|_2 \right).$$

Since the attention with Pre-Norm and QK-Norm lies between Pre-Norm and QKV-Norm, its proof can be directly derived from those of the other two. Therefore, we defer its proof to the end. As for the attention with QKV-Norm, we have

$$\left\| \frac{\partial S_N}{\partial W_O} \right\|_F = \|A_N V_N \otimes I_d\|_F = \|A_N V_N\|_F \|I_d\|_F \leq \sqrt{d} \|A_N\|_2 \|V_N\|_F \leq s\sqrt{d}.$$

For $\frac{\partial S_N}{\partial W_V}$, we have

$$\left\| \frac{\partial S_N}{\partial W_V} \right\|_F = \left\| (A \otimes W_O^\top) \frac{\partial V_N}{\partial V} (X \otimes I_{d_k}) \right\|_F$$

$$\leq \left\| A \otimes W_O^\top \right\|_2 \left\| \frac{\partial V_N}{\partial V} (X \otimes I_{d_k}) \right\|_F$$

$$\leq \sqrt{s} \|W_O\|_2 \left\| \frac{\partial V_N}{\partial V} (X \otimes I_{d_k}) \right\|_F.$$

According to Eq. (21), we get

$$\left\| \frac{\partial V_N}{\partial V} (X \otimes I_{d_k}) \right\|_F = \left\| \text{blockdiag} \left( \frac{\sqrt{d_k}}{\|V_{i,:}\|_2} \left( I_{d_k} - \frac{V_{i,:} V_{i,:}^\top}{\|V_{i,:}\|_2^2} \right) \right) (X \otimes I_{d_k}) \right\|_F$$

$$= \left\| \begin{bmatrix} X_{1,:}^\top \otimes \left( \frac{\sqrt{d_k}}{\|V_{1,:}\|_2} \left( I_{d_k} - \frac{V_{1,:} V_{1,:}^\top}{\|V_{1,:}\|_2^2} \right) \right) \\ \vdots \\ X_{s,:}^\top \otimes \left( \frac{\sqrt{d_k}}{\|V_{s,:}\|_2} \left( I_{d_k} - \frac{V_{s,:} V_{s,:}^\top}{\|V_{s,:}\|_2^2} \right) \right) \end{bmatrix} \right\|_F$$

$$= \sqrt{d_k(d_k - 1) \sum_{i=1}^{s} \frac{\|X_{i,:}\|_2^2}{\|V_{i,:}\|_2^2}}$$

$$= \sqrt{d_k(d_k - 1) \sum_{i=1}^{s} \frac{\|X_{i,:}\|_2^2}{\|W_V^\top X_{i,:}\|_2^2}}$$

$$\leq \frac{\sqrt{s} d_k}{\sigma_{\min}^V}.$$

Similarly, we can get

$$\left\| \frac{\partial Q_N}{\partial Q} (X \otimes I_{d_k}) \right\|_F \leq \frac{\sqrt{s} d_k}{\sigma_{\min}^Q}, \quad \left\| \frac{\partial K_N}{\partial K} (X \otimes I_{d_k}) \right\|_F \leq \frac{\sqrt{s} d_k}{\sigma_{\min}^K}.$$

It follows that

$$\left\| \frac{\partial S_N}{\partial W_V} \right\|_F \leq \frac{s d_k}{\sigma_{\min}^V} \|W_O\|_2. \tag{42}$$

For $\frac{\partial S_N}{\partial W_Q}$, we have

$$\left\| \frac{\partial S_N}{\partial W_Q} \right\|_F = \left\| (I_s \otimes W_O^\top V_N^\top) \frac{\partial A_N}{\partial M_N} \left( \frac{I_s \otimes K_N}{\sqrt{d_k}} \right) \frac{\partial Q_N}{\partial Q} (X \otimes I_{d_k}) \right\|_F$$

$$\leq \frac{1}{\sqrt{d_k}} \left\| (I_s \otimes W_O^\top V_N^\top) \frac{\partial A_N}{\partial M_N} (I_s \otimes K_N) \right\|_2 \left\| \frac{\partial Q_N}{\partial Q} (X \otimes I_{d_k}) \right\|_F$$

$$\leq \frac{1}{\sqrt{d_k}} \left\| (I_s \otimes W_O^\top V_N^\top) \text{blockdiag}(\text{diag}((A_N)_{i,:}) - (A_N)_{i,:}(A_N)_{i,:}^\top) (I_s \otimes K_N) \right\|_2 \frac{\sqrt{s} d_k}{\sigma_{\min}^Q}$$

$$= \frac{\sqrt{s} d_k}{\sigma_{\min}^Q} \left\| \text{blockdiag}(W_O^\top V_N^\top (\text{diag}((A_N)_{i,:}) - (A_N)_{i,:}(A_N)_{i,:}^\top) K_N) \right\|_2$$

$$\leq \frac{\sqrt{s} d_k}{\sigma_{\min}^Q} \|W_O\|_2 \|V_N\|_2 \frac{1}{2} \|K_N\|_2$$

$$\leq \frac{s \sqrt{s} d_k}{2 \sigma_{\min}^Q} \|W_O\|_2.$$

Similarly, for $\frac{\partial S_N}{\partial W_K}$, we have

$$\left\| \frac{\partial S_N}{\partial W_K} \right\|_F \leq \frac{s \sqrt{s} d_k}{2 \sigma_{\min}^K} \|W_O\|_2.$$

Therefore,

$$\left\| \frac{\partial S_N}{\partial W_O} \right\|_F = \mathcal{O}\left( s\sqrt{d} \right),$$

$$\left\| \frac{\partial S_N}{\partial W_V} \right\|_F = \mathcal{O}\left( \frac{sd_k}{\sigma_{\min}^V} \|W_O\|_2 \right),$$

$$\left\| \frac{\partial S_N}{\partial W_Q} \right\|_F = \mathcal{O}\left( \frac{s\sqrt{sd_k}}{\sigma_{\min}^Q} \|W_O\|_2 \right),$$

$$\left\| \frac{\partial S_N}{\partial W_K} \right\|_F = \mathcal{O}\left( \frac{s\sqrt{sd_k}}{\sigma_{\min}^K} \|W_O\|_2 \right).$$

Finally, we present the proof for the attention mechanism with Pre-Norm and QK-Norm. For $\frac{\partial \hat{S}}{\partial W_O}$ and $\frac{\partial \hat{S}}{\partial W_O}$, whose proofs are essentially identical to that of Pre-Norm, we have

$$\left\| \frac{\partial \hat{S}}{\partial W_O} \right\|_F = \left\| \hat{A}_N X_N W_V \otimes I_d \right\|_F = \|\hat{A}_N X_N W_V\|_F \|I_d\|_F \leq \sqrt{d}\|\hat{A}_N\|_2 \|X_N\|_F \|W_V\|_2 \leq s\sqrt{d}\|W_V\|_2,$$

$$\left\| \frac{\partial \hat{S}}{\partial W_V} \right\|_F = \left\| \hat{A}_N X_N \otimes W_O^\top \right\|_F = \|\hat{A}_N X_N\|_F \|W_O\|_F \leq \|\hat{A}_N\|_2 \|X_N\|_F \|W_O\|_F \leq s\|W_O\|_F.$$

For $\frac{\partial \hat{S}}{\partial W_Q}$ and $\frac{\partial \hat{S}}{\partial W_K}$, whose proofs are similar to that of QKV-Norm, we have

$$\left\| \frac{\partial \hat{S}}{\partial W_Q} \right\|_F = \left\| (I_s \otimes W_O^\top W_V^\top X_N^\top) \frac{\partial \hat{A}_N}{\partial \hat{M}_N} \left( \frac{I_s \otimes \hat{K}_N}{d_k} \right) \frac{\partial \hat{Q}_N}{\partial \hat{Q}} (X_N \otimes I_{d_k}) \right\|_F$$

$$\leq \frac{1}{\sqrt{d_k}} \left\| (I_s \otimes W_O^\top W_V^\top X_N^\top) \frac{\partial \hat{A}_N}{\partial \hat{M}_N} (I_s \otimes \hat{K}_N) \right\|_2 \left\| \frac{\partial \hat{Q}_N}{\partial \hat{Q}} (X_N \otimes I_{d_k}) \right\|_F$$

$$\leq \frac{1}{\sqrt{d_k}} \left\| (I_s \otimes W_O^\top W_V^\top X_N^\top) \, \text{blockdiag}(\text{diag}((\hat{A}_N)_{i,:}) - (\hat{A}_N)_{i,:}(\hat{A}_N)_{i,:}^\top) \left( I_s \otimes \hat{K}_N \right) \right\|_2 \frac{\sqrt{sd_k}}{\sigma_{\min}^Q}$$

$$= \frac{\sqrt{sd_k}}{\sigma_{\min}^Q} \left\| \text{blockdiag}(W_O^\top W_V^\top X_N^\top (\text{diag}((\hat{A}_N)_{i,:}) - (\hat{A}_N)_{i,:}(\hat{A}_N)_{i,:}^\top) \hat{K}_N) \right\|_2$$

$$\leq \frac{\sqrt{sd_k}}{\sigma_{\min}^Q} \|W_O\|_2 \|W_V\|_2 \|X_N\|_2 \frac{1}{2} \|\hat{K}_N\|_2$$

$$\leq \frac{\sqrt{sd_k}}{\sigma_{\min}^Q} \|W_O\|_2 \|W_V\|_2 \sqrt{s} \frac{1}{2} \sqrt{s}$$

$$\leq \frac{s\sqrt{sd_k}}{2\sigma_{\min}^Q} \|W_V\|_2 \|W_O\|_2.$$

Similarly,

$$\left\| \frac{\partial \hat{S}}{\partial W_K} \right\|_F \leq \frac{s\sqrt{sd_k}}{2\sigma_{\min}^Q} \|W_V\|_2 \|W_O\|_2.$$

Hence,

$$\left\| \frac{\partial \hat{S}}{\partial W_O} \right\|_F = \mathcal{O}\left( s\sqrt{d}\|W_V\|_2 \right),$$

$$\left\| \frac{\partial \hat{S}}{\partial W_V} \right\|_F = \mathcal{O}\left( s\|W_O\|_F \right),$$

$$\left\| \frac{\partial \hat{S}}{\partial W_Q} \right\|_F = \mathcal{O}\left( \frac{s\sqrt{sd_k}}{\sigma_{\min}^Q} \|W_V\|_2 \|W_O\|_2 \right),$$

$$\left\| \frac{\partial \hat{S}}{\partial W_K} \right\|_F = \mathcal{O}\left( \frac{s\sqrt{sd_k}}{\sigma_{\min}^K} \|W_V\|_2 \|W_O\|_2 \right).$$

$\square$

## B.5  Expand Theoretical Clarifications to LayerNorm

In the following, we demonstrate that RMSNorm and LayerNorm exhibit no fundamental differences, both in theoretical analysis and empirical observations. It is worth noting that, given the vast majority of popular LLMs are based on RMSNorm, our experiments and conclusions are broadly applicable to standard LLM architectures.

Suppose the input is given by $X \in \mathbb{R}^{s \times d}$. Let $P = I - \frac{1}{d}1_d 1_d^\top$ . Then

$$\mathbb{E}X = X\frac{1}{d}1_d 1_d^\top, \text{ and } \quad X - \mathbb{E}X = X - X\frac{1}{d}1_d 1_d^\top = XP.$$

For simplicity, we omit the affine transformation in the normalization layers. It follows that

$$\text{LayerNorm}(X) = \frac{X - \mathbb{E}X}{Var(X)} = \frac{X - \mathbb{E}X}{\text{RMS}(X - \mathbb{E}X)} = \frac{XP}{\text{RMS}(XP)} = \text{RMSNorm}(XP).$$

Hence, analogous to the gradient of RMSNorm (Eq. 20), the gradient of LayerNorm is given by

$$\begin{aligned}
\frac{\partial \text{LayerNorm}(X)}{\partial X} &= \frac{\partial \text{RMSNorm}(XP)}{\partial X} \\
&= \text{blockdiag}\left( \frac{\partial \text{RMSNorm}((XP)_i)}{\partial X_i} \right) \\
&= \text{blockdiag}\left( \frac{\sqrt{d}}{\|(XP)_i\|_2} \left( I_d - \frac{(XP)_i(XP)_i^\top}{\|(XP)_i\|_2^2} \right) P \right),
\end{aligned}$$

where $X_i$ means that the i-th column of $X$. Note that $P$ is a positive semidefinite matrix with eigenvalues bounded between 0 and 1, hence $\|P\|_2 = 1$. As a result, the gradient norms of LayerNorm and RMSNorm differ only by a constant factor. This implies that the main result, Theorem 2, also holds for LayerNorm.

On the experimental side, we conducted controlled comparison using models of identical size (550M parameters), each trained on 400B tokens. Both models adopt the Pre-Norm architecture and employ the Megatron initialization scheme; the only difference lies in the normalization method—one uses RMSNorm, while the other uses LayerNorm. As shown in Table 8, the training losses of RMSNorm and LayerNorm are nearly identical, with a marginal difference of only 0.0008. In the context of large-scale models, a loss difference smaller than 0.001 is typically considered negligible.

Table 8: Training loss comparison between RMSNorm and LayerNorm under identical training settings.

| Methods | Training Loss |
|---------|---------------|
| RMSNorm | 2.7631 |
| LayerNorm | 2.7639 |

# C Details of Experiments

## C.1 PyTorch Style Implementation of HybridNorm

We provide a PyTorch-style implementation of HybridNorm below, and a more detailed implementation can be found at `https://github.com/BryceZhuo/HybridNorm`.

---

**Algorithm 1** PyTorch style pseudocode for a Transformer block with HybridNorm

---

```python
# q_norm, k_norm, v_norm, ffn_norm are normalization layers
# attn_proj and attn_out are linear layers
# attn is the attention
# ffn is the feedforward network

def forward(x):
    # Attention block
    res = x                                          # shape (b, s, d)
    q, k, v = attn_proj(x).split((d, d, d), dim=-1)  # shape (b, s, d)
    # dk = d / h, h is the number of attention heads
    q, k, v = q.view(b, s, h, dk)
    q, k, v = q_norm(q), k_norm(k), v_norm(v)
    x = attn(q,k,v)                                  # shape (b, s, d)
    x = attn_out(x) + res

    # FFN block
    x = ffn_norm(x)
    x = ffn(x) + x

    return x
```

---

## C.2 Architectures of Different Models

For dense models, we adopt a decoder-only transformer architecture akin to LLaMA 3.2 [14], with model sizes ranging from 151M to 1.2B parameters. For the MoE model, we follow the structure of OLMoE [15]. The specific architecture of models is summarized in Table 9. All experiments are conducted on NVIDIA A100-80G GPUs, utilizing 32 GPUs for dense models with fewer than 1B parameters, and 64 GPUs for the 1.2B dense model and MoE-1B-7B. Pretraining durations vary from one to ten days, depending on the model size and the number of training tokens.

Table 9: Model architecture for dense models and MoE models.

|  | Dense-151M | Dense-285M | Dense-550M | Dense-543M | Dense-1.2B | MoE-1B-7B |
|---|---|---|---|---|---|---|
| Model Dimension | 768 | 1024 | 1536 | 1024 | 2048 | 2048 |
| FFN Dimension | 2048 | 4096 | 4096 | 4096 | 9192 | 1024 |
| Attention heads | 16 | 16 | 16 | 16 | 32 | 16 |
| Key/Value Heads | 4 | 4 | 4 | 4 | 8 | 16 |
| Layers | 12 | 12 | 16 | 29 | 16 | 16 |
| Vocabulary Size | 100278 | 100278 | 100278 | 100278 | 100278 | 50280 |
| Weight Tying | True | True | True | True | True | False |
| Context Length | 4096 | 4096 | 4096 | 4096 | 4096 | 4096 |
| Expert Granularity | - | - | - | - | - | 8 in 64 |

## C.3 Hyperparameters for Pretraining

For the training hyperparameters, we primarily adopt the configuration outlined in OLMo 2 [30] and OLMoE [15]. The training hyperparameters for our models across different sizes are presented in Table 10. The model is trained using the AdamW optimizer with a learning rate (LR) of 3e-4 (4e-4 for MoE), which is scheduled to decay following a cosine function. The minimum LR

Table 10: Hyperparameters for Pretraining.

|  | Dense Model | MoE-1B-7B |
|---|---|---|
| Optimizer | AdamW | AdamW |
| Learning Rate (LR) | 3e-4 | 4e-4 |
| Minimum LR | 3e-5 | 5e-5 |
| LR Schedule | cosine | cosine |
| Weight Decay | 0.1 | 0.1 |
| $\beta_1$ | 0.9 | 0.9 |
| $\beta_2$ | 0.95 | 0.95 |
| Gradient Clipping | 1 | 1 |
| Batch Size | 1024 | 1024 |
| Warmup Tokens | 8,388,608,000 | 10,485,760,000 |
| Init Distribution | Megatron | Megatron |
| Init std | $1/\sqrt{2.5d}$ | $1/\sqrt{2.5d}$ |
| Init Truncation | 3 std | 3 std |
| RoPE $\theta$ | 500000 | 10000 |
| Activation | SwiGLU | SwiGLU |
| Load Balancing Loss Weight | - | 0.01 |
| Router z-loss Weight | - | 0.001 |

is set to 3e-5 (5e-5 for MoE) to prevent excessively small updates in the later stages of training. A weight decay of 0.1 is applied to regularize the model and prevent overfitting. The AdamW optimizer employs $\beta_1 = 0.9$ and $\beta_2 = 0.95$ to control the first and second momentum estimates, respectively. Gradient clipping is utilized with a threshold of 1 to mitigate the impact of large gradients during optimization. The model's training also incorporates a warmup phase with a total of 8,388,608,000 tokens (10,485,760,000 for MoE). The initialization of the model's parameters follows a normal distribution with a standard deviation defined as $1/\sqrt{2.5d}$, where $d$ is the model dimension. Furthermore, the initialization is truncated at 3 standard deviations to ensure a more stable starting point for training. The RoPE (Rotary Position Embedding) parameter $\theta$ is set to 500,000 (10000 for MoE), controlling the scale of position encodings. Finally, the activation function used in the model is SwiGLU, which has been shown to outperform traditional activation functions in various tasks.

## D    Computational Overhead

The direct contribution of RMSNorm to the overall parameter count and computational cost of a large Transformer is, in fact, negligible. Below, we provide a detailed analysis to support this claim.

For simplicity, we consider only the MHA and the SwiGLU activation function, excluding the Embedding and Output layers. Suppose the hidden dimension is $d$, the intermediate FFN size is $\frac{8}{3}d$, the number of layers is $L$, and sequence length is $s$. As shown in the table below, RMSNorm constitutes only a negligible fraction of the actual runtime and memory consumption in Transformer under both Pre-Norm and HybridNorm configurations. And the ratio decreases as the model size increases. Moreover, when employing GQA, the relative overhead of RMSNorm in HybridNorm is further diminished.

**Parameters** Each RMSNorm layer introduces $d$ parameters. In a Pre-Norm architecture, there are two RMSNorm layers per Transformer layer, resulting in a total of $2dL$ parameters. In HybridNorm, there are four RMSNorm layers per Transformer layer, yielding $4dL$ parameters in total. Each attention block has four weight matrices (for Q, K, V, and O projections), thereby contributing $4d^2L$ parameters in total for the MHA component. Similarly, the FFN component introduces an additional $8d^2L$ parameters.

**Computation** The FLOPs for RMSNorm to process a single token vector of dimension $d$ is $(4d+4)$. Since the constant term becomes negligible for any reasonably large $d$, this can be simplified to $4d$. Accordingly, for Pre-Norm, the $2L$ RMSNorm operations incur a total cost of approximately $8sdL$,

Table 11: Parameter and computation cost comparison between Pre-Norm and HybridNorm architectures. $L$ denotes the number of layers, $d$ the hidden dimension, and $s$ the sequence length.

| Type | Architecture | RMSNorm | Main Transformer Components (MHA & FFN) | Ratio (RMSNorm / Total) |
|---|---|---|---|---|
| **Parameter** | Pre-Norm | $2dL$ | $12d^2L$ | $\dfrac{2dL}{12d^2L + 2dL} \approx \dfrac{1}{6d}$ |
| | HybridNorm | $4dL$ | $12d^2L$ | $\dfrac{4dL}{12d^2L + 4dL} \approx \dfrac{1}{3d}$ |
| **Computation (FLOPs)** | Pre-Norm | $8sdL$ | $(24sd^2 + 4s^2d)L$ | $\dfrac{8sdL}{(24sd^2 + 4s^2d)L} = \dfrac{2}{6d + s}$ |
| | HybridNorm | $16sdL$ | $(24sd^2 + 4s^2d)L$ | $\dfrac{16sdL}{(24sd^2 + 4s^2d)L} = \dfrac{4}{6d + s}$ |

whereas HybridNorm incurs $16sdL$. In practice, however, the dominant computational overhead in a Transformer stems from matrix-vector multiplications, primarily within MHA and FFN modules. Specifically, the FLOPs for MHA amount to $(8sd^2 + 4s^2d)L$, while the FLOPs for FFN total $16sd^2L$.

# E  Additional Experimental Results

## E.1  Comparison with Other Methods

We compare the downstream evaluation results of 1.2B dense models trained on 200B tokens using five normalization strategies: Post-Norm, Pre-Norm, Mix-LN, HybridNorm, and HybridNorm$^*$. As shown in Table 12, **HybridNorm$^*$ consistently delivers the highest average performance across eight downstream tasks**, outperforming all other methods both on average and in the majority of individual cases.

In particular, HybridNorm$^*$ achieves the top scores on HellaSwag (59.56), SciQ (90.70), ARC-C (36.15), PIQA (73.83, and COPA (80.40), while remaining competitive on the remaining tasks. In contrast to Post-Norm and Pre-Norm, which show strong results on select benchmarks but suffer from variability elsewhere, HybridNorm$^*$ exhibits **robust and consistently balanced performance**. Notably, it attains an average score of 60.67, surpassing the second-best method (Pre-Norm, 59.56) by over one point, underscoring its effectiveness in enhancing generalization across a diverse set of evaluation tasks.

We also observe that Post-Norm underperforms Pre-Norm in both Table 12 and Table 6. This can be attributed to the fact that, although Post-Norm generally offers a higher performance upper bound, it suffers from reduced training stability and requires extensive hyperparameter tuning [4, 6, 40]. In our experiments, we adopt the default hyperparameters from OLMo 2 without performing extensive hyperparameter search; hence, it is unsurprising that Post-Norm does not outperform Pre-Norm under these settings. This observation motivates us to combine the higher performance ceiling of Post-Norm with the superior training stability and hyperparameter robustness of Pre-Norm. Our experimental results show that HybridNorm achieves stronger robustness to hyperparameters, enabling a balanced trade-off between stability and performance without the need for extensive tuning.

We also have extended Table 6 to include DeepNorm [7], Sandwich-LN [43], and OutputNorm (OLMo 2 [30]), as shown in Table 13. We have also clarified in the revised manuscript that OutputNorm refers to the normalization strategy employed in OLMo 2, where RMSNorm is applied to the outputs of attention and MLP sublayers.

It shows that HybridNorm consistently outperforms all other normalization strategies, including the aforementioned advanced methods, across all metrics. This highlights the strength of our approach in both generative perplexity tasks and HellaSwag.

Table 12: Downstream evaluation results of 1.2B dense models with Post-Norm, Pre-Norm, Mix-LN, HybridNorm, and HybridNorm* under 200B training tokens. OQA refers to OpenbookQA.

| Methods | BasicArithmetic | HellaSwag | SciQ | ARC-C | ARC-E | PIQA | OQA | COPA | Avg.↑ |
|---|---|---|---|---|---|---|---|---|---|
| Post-Norm | 37.07 | 57.44 | 88.86 | 35.85 | 66.32 | 73.23 | **37.62** | 75.80 | 59.02 |
| Pre-Norm | **40.06** | 57.38 | 89.78 | 32.14 | 67.12 | 73.04 | 36.58 | 80.40 | 59.56 |
| Mix-LN | 33.23 | 56.91 | 89.20 | 33.78 | **69.30** | 72.25 | 37.00 | 79.00 | 58.83 |
| HybridNorm | 35.29 | 58.24 | 88.36 | 35.69 | 68.63 | 73.45 | 37.02 | 76.80 | 59.18 |
| HybridNorm* | 39.46 | **59.56** | **90.70** | **36.15** | 68.25 | **73.83** | 37.00 | **80.40** | **60.67** |

Table 13: Comparison of normalization methods on 550M models.

| Methods | Training Loss↓ | C4 PPL↓ | Pile PPL↓ | HellaSwag↑ |
|---|---|---|---|---|
| Post-Norm | 2.760 | 20.43 | 10.57 | 51.20 |
| Pre-Norm | 2.751 | 20.30 | 10.48 | 51.97 |
| Mix-LN | 2.760 | 20.43 | 10.56 | 51.29 |
| DeepNorm | 2.746 | 20.34 | 10.48 | 52.11 |
| Sandwich-LN | 2.751 | 20.45 | 10.58 | 52.07 |
| OutputNorm (OLMo 2) | 2.750 | 20.34 | 10.44 | 52.82 |
| HybridNorm | 2.737 | 20.00 | 10.29 | 53.35 |
| HybridNorm* | **2.731** | **19.85** | **10.25** | **53.36** |

## E.2 Signal Propagation

Following [45], we plotted the evolution of the cosine similarity between tokens during pretraining. As shown in Figure 7, both Pre-Norm and Mix-LN exhibit a notable increase in token similarity in certain layers, indicating a tendency toward representation degeneration. In contrast, HybridNorm maintains consistently lower similarity across most layers, suggesting better capabilities in information representation [45]. This highlights the benefit of employing QKV-Norm in the attention module and Post-Norm in FFN to improve information flow.

## E.3 Entropy Dynamics

Following [49], we also examine the layerwise entropy evolution during pre-training. From Figure 8, HybridNorm maintains entropy within a relatively stable range, avoiding the sharp fluctuations observed in other methods. A stable entropy distribution is known to correlate with smoother training dynamics and improved optimization stability [49], providing further empirical support for the robustness of HybridNorm.

## E.4 Overall Results for Dense Models

Overall results for the dense model are presented in Figure 9, depicting validation losses and downstream evaluations over 1T training tokens. The comparison includes models with Pre-Norm, HybridNorm, and HybridNorm*. One can see that both HybridNorm and HybridNorm* outperform Pre-Norm, with HybridNorm* achieving the lowest training and validation losses while delivering the best downstream performance on average.

## E.5 Overall Results for MoE Models

Overall results for the MoE model are presented in Figure 10, illustrating validation losses and downstream evaluations over 500 billion training tokens. The comparison focuses on models employing Pre-Norm and HybridNorm*. From the figures, we can see that HybridNorm* achieves lower training loss and validation loss compared to Pre-Norm on all datasets, such as C4, Books, and Pile. Additionally, HybridNorm* outperforms Pre-Norm on most downstream tasks, though there

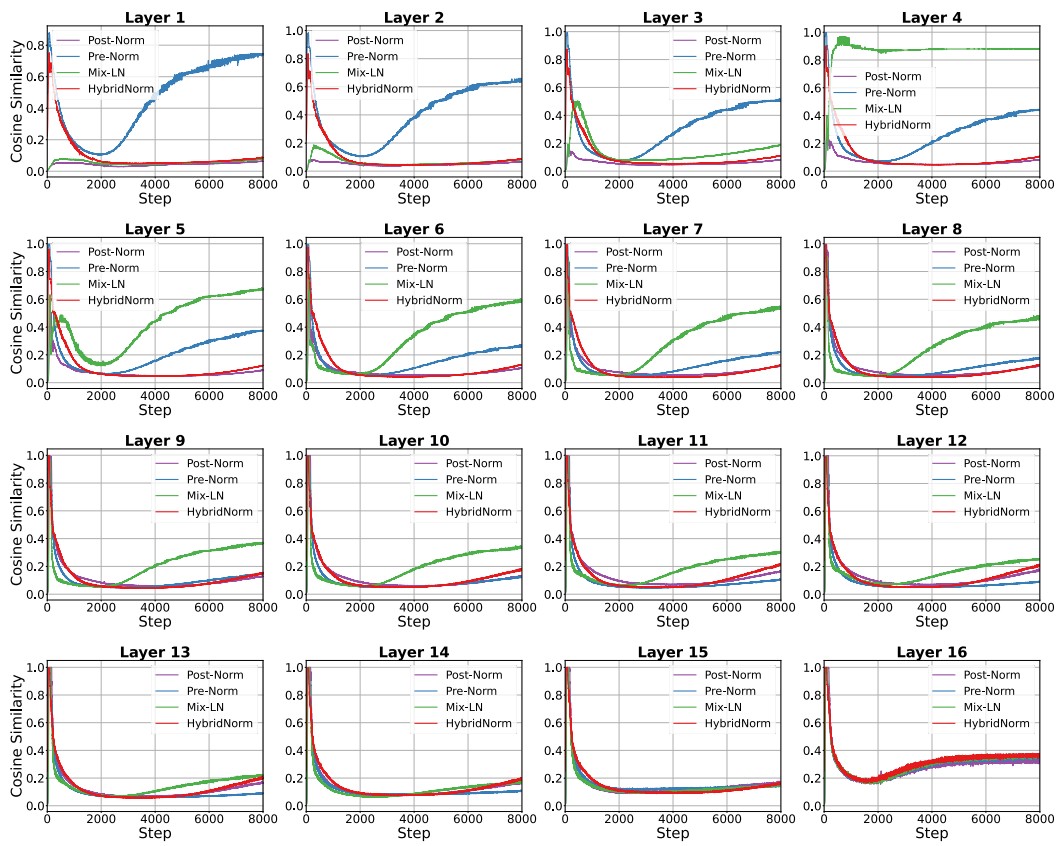

Figure 7: Evolution of the cosine similarity between tokens for Post-Norm, Pre-Norm, Mix-LN, and HybridNorm at different layers.

are some cases where it underperforms. On average, however, HybridNorm* demonstrates superior downstream performance.

# F   Essential Differences Between Pre-Norm and Post-Norm

To facilitate a unified analysis of the various normalization variants, we propose a categorization of Pre-Norm and Post-Norm based on their distinct approaches to residual connections (refer to Figure 11, particularly the sections highlighted by the dashed boxes). From this perspective, the FFN sublayer in HybridNorm can be considered as adopting a connection scheme similar to the Post-Norm.

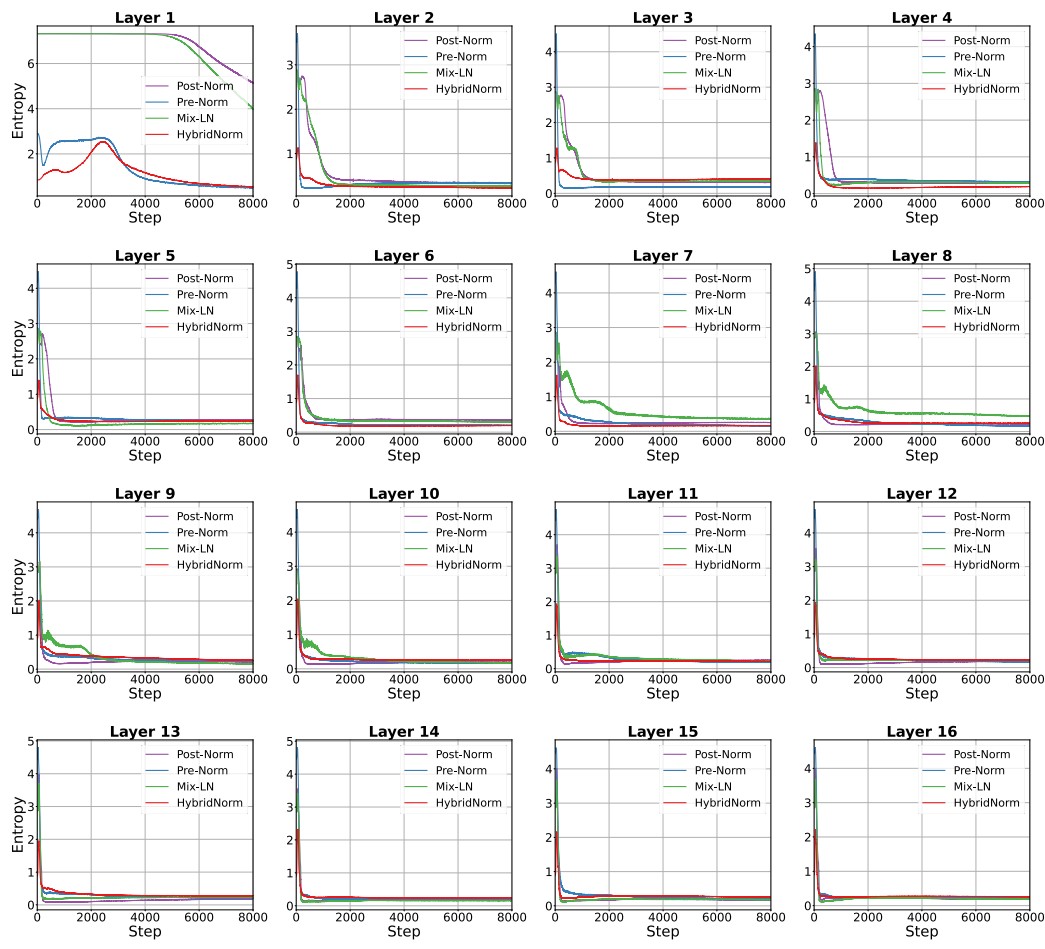

Figure 8: Evolution of the layerwise entropy for Post-Norm, Pre-Norm, Mix-LN, and HybridNorm at different layers.

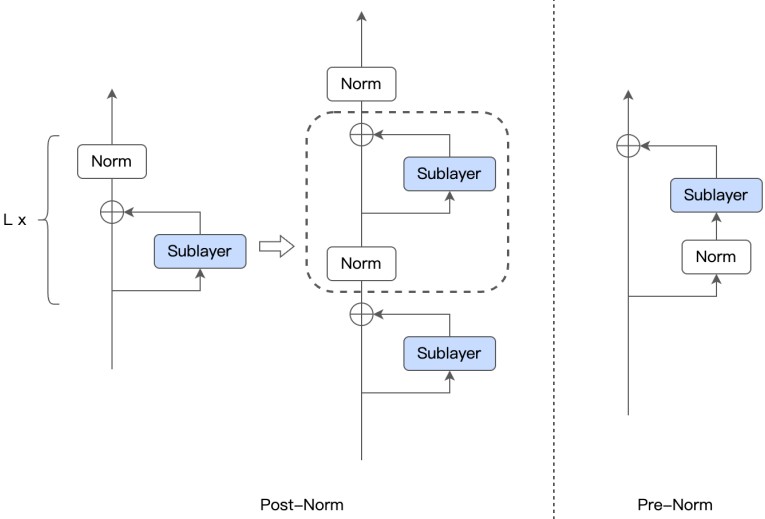

Figure 11: A unified view of Pre-Norm and Post-Norm

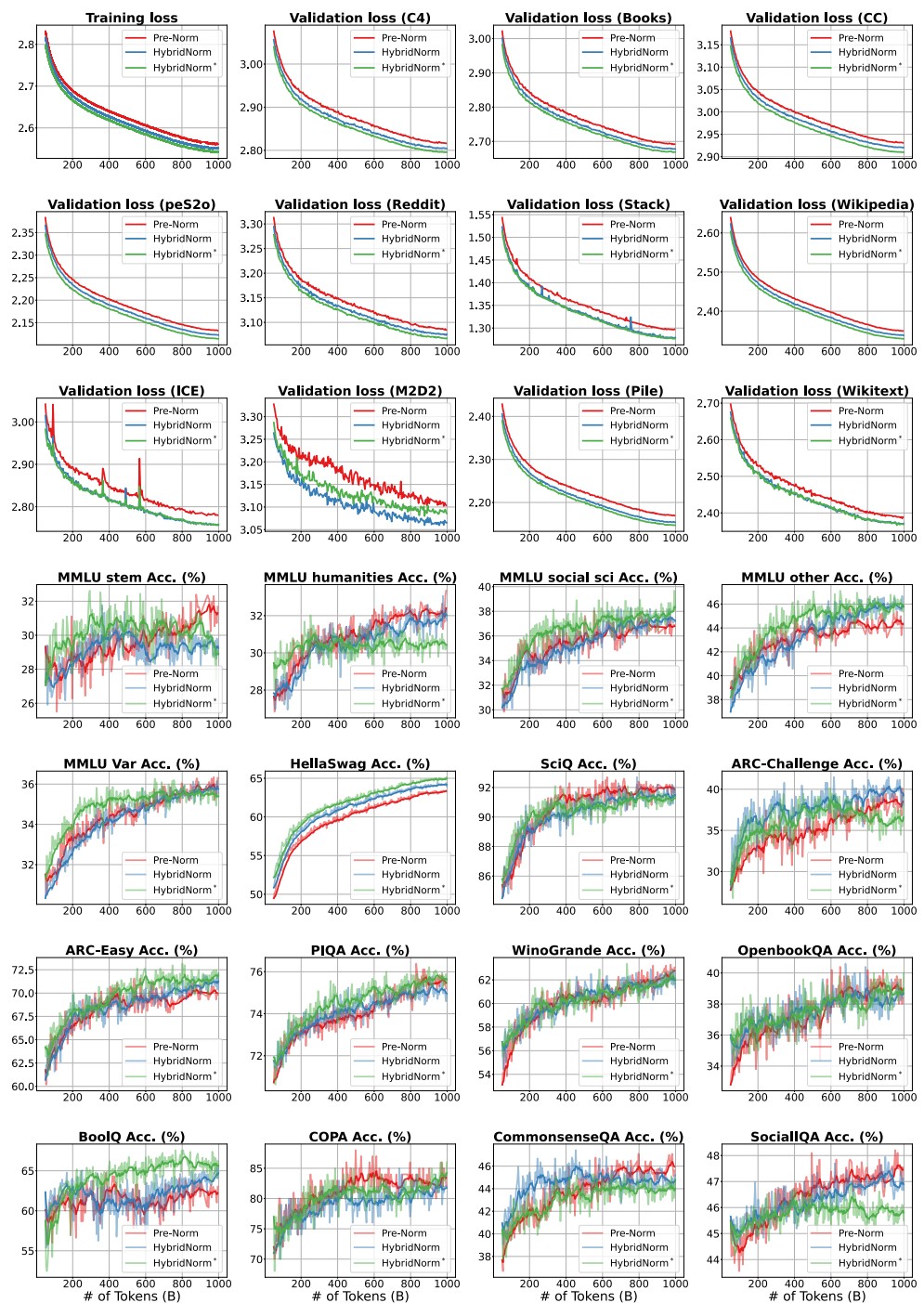

Figure 9: Overall loss and downstream evaluations for the 1.2B dense models with 1T training tokens.

# G   Formulas for Different Positions of Normalization Layers

In this section, we present the mathematical formulations for various normalization techniques. We begin by introducing the normalization layer within the attention mechanism.

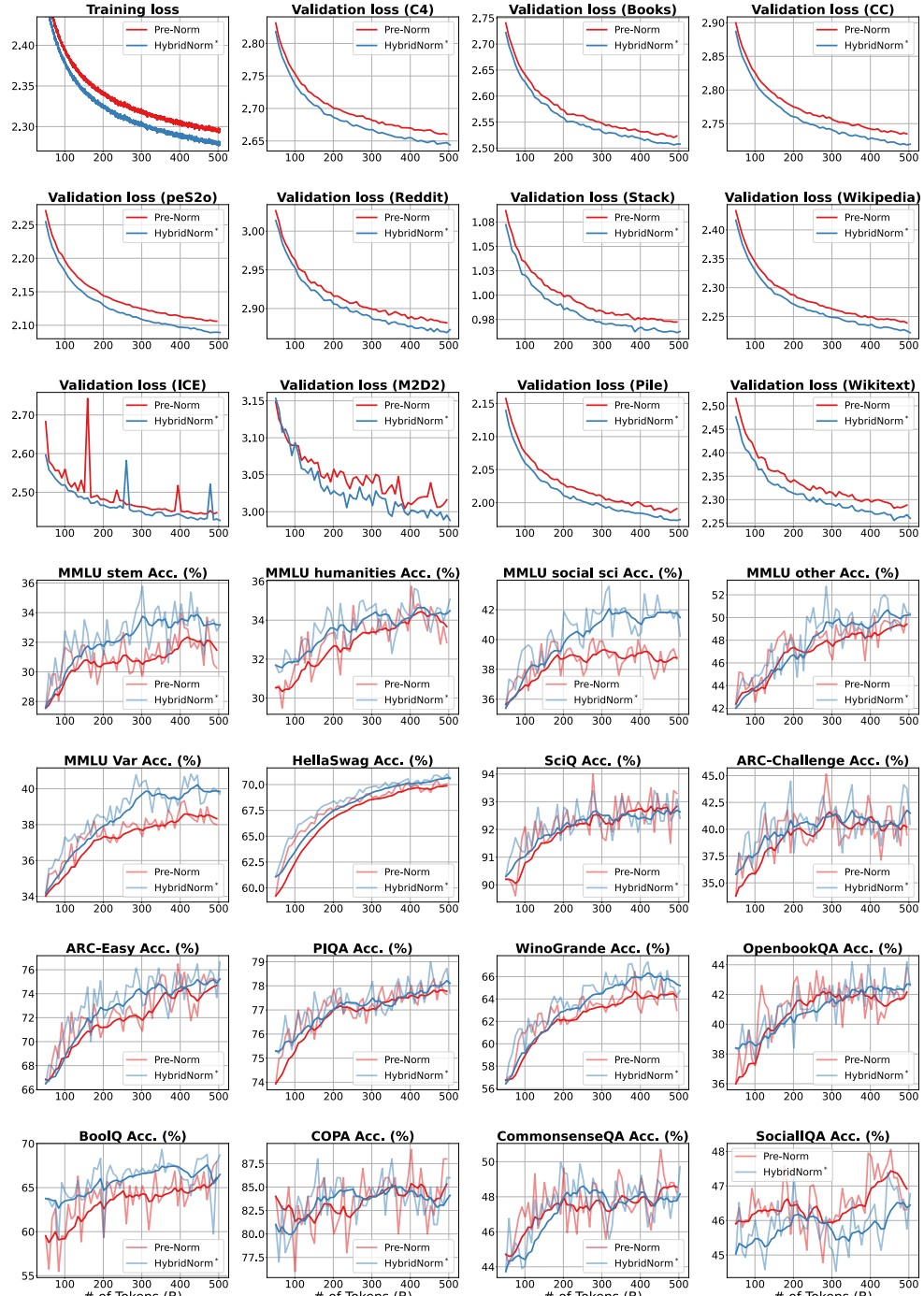

Figure 10: Overall loss and downstream evaluations for the MoE models with 500B training tokens.

**Vanilla scaled dot-product attention** are show in Eq. 1, and **attention with QKV-Norm** is defined in Eq. 5. Similarly, **attention with QK-Norm** is defined as

$$\text{attn}_{QK}(Q, K, V) = \text{softmax}\left(\frac{\text{Norm}(Q)\text{Norm}(K)^\top}{\sqrt{d_k}}\right) V. \tag{43}$$

**Attention with KV-Norm** is defined as

$$\text{attn}_{KV}(Q, K, V) = \text{softmax}\left(\frac{Q\text{Norm}(K)^\top}{\sqrt{d_k}}\right)\text{Norm}(V). \tag{44}$$

As mentioned in Section 5.4, we extend traditional normalization approaches by considering not only the Query (Q), Key (K), and Value (V) components but also the Context (C), which refers to the output of the attention mechanism. And **attention with QKVC-Norm** is defined as

$$\text{attn}_{QKVC}(Q, K, V) = \text{Norm}\left(\text{softmax}\left(\frac{\text{Norm}(Q)\text{Norm}(K)^\top}{\sqrt{d_k}}\right)\text{Norm}(V)\right). \tag{45}$$

**Attention with QKC-Norm** is defined as

$$\text{attn}_{QKC}(Q, K, V) = \text{Norm}\left(\text{softmax}\left(\frac{\text{Norm}(Q)\text{Norm}(K)^\top}{\sqrt{d_k}}\right)V\right). \tag{46}$$

**Attention with KC-Norm** is defined as

$$\text{attn}_{KC}(Q, K, V) = \text{Norm}\left(\text{softmax}\left(\frac{Q\text{Norm}(K)^\top}{\sqrt{d_k}}\right)V\right). \tag{47}$$

Then we denote MHA with $\text{attn}_\#$ as $\text{MHA}_\#$ for $\# \in \{QKVC, QKV, QKC, QK, KV, KC\}$,

$$\text{MHA}(X)_\# = \text{Concat}(\text{head}_1, \ldots, \text{head}_h)W^O, \tag{48}$$

where $\text{head}_i = \text{attn}_\#(Q_i, K_i, V_i)$ for $i = 1, 2, \ldots, h$, $\{\bullet_i\}_{i=1}^h = \text{Split}(XW_\bullet)$ for $\bullet \in \{Q, K, V\}$, and $W_Q, W_K, W_V, W_O \in \mathbb{R}^{d \times d}$ are learnable parameters.

With the aforementioned definitions in hand, we present the mathematical formulations for the methods discussed in the Ablation Study below ($\# \in \{QKVC, QKV, QKC, QK, KV, KC\}$).

**#-Post**:

$$Y^l = \text{MHA}_\#(X^l) + X^l, \tag{49}$$

$$X^{l+1} = \text{FFN}(\text{Norm}(Y^l)) + \text{Norm}(Y^l). \tag{50}$$

**#-Pre**:

$$Y^l = \text{MHA}_\#(X^l) + X^l, \tag{51}$$

$$X^{l+1} = \text{FFN}(\text{Norm}(Y^l)) + Y^l. \tag{52}$$

**Pre-#-Post**:

$$Y^l = \text{MHA}_\#(\text{Norm}(X^l)) + X^l, \tag{53}$$

$$X^{l+1} = \text{FFN}(\text{Norm}(Y^l)) + \text{Norm}(Y^l). \tag{54}$$

**Pre-#-Pre**:

$$Y^l = \text{MHA}_\#(\text{Norm}(X^l)) + X^l, \tag{55}$$

$$X^{l+1} = \text{FFN}(\text{Norm}(Y^l)) + Y^l. \tag{56}$$

**Pre-Post**:

$$Y^l = \text{MHA}(\text{Norm}(X^l)) + X^l, \tag{57}$$

$$X^{l+1} = \text{FFN}(\text{Norm}(Y^l)) + \text{Norm}(Y^l). \tag{58}$$

**Post-Pre**:

$$Y^l = \text{MHA}(\text{Norm}(X^l)) + \text{Norm}(X^l), \tag{59}$$

$$X^{l+1} = \text{FFN}(\text{Norm}(Y^l)) + Y^l. \tag{60}$$

# H Broader Impacts and Limitations

## H.1 Broader Impacts

This paper proposes HybridNorm, a simple yet effective hybrid normalization technique that improves the training stability and performance of transformers. It has the potential to assist the LLM community in advancing transformer architectures and enhancing their overall effectiveness. While there may be societal implications of our work, none of which we feel must be specifically highlighted here.

## H.2 Limitations

First, due to limited computational resources, our experiments are conducted on models ranging from 151M to 7B parameters. While our method shows strong effectiveness on smaller models, its performance on larger-scale models has not yet been empirically validated. Second, although our theoretical analysis demonstrates improved gradient stability, this does not directly guarantee better overall model performance.

