# OpenReview forum: "HybridNorm: Towards Stable and Efficient Transformer Training via Hybrid Normalization"
_NeurIPS.cc/2025/Conference — NeurIPS 2025 poster_

### Official Review · Reviewer_zMv1 · 2025-07-02

**Clarity:** 3
**Significance:** 2
**Originality:** 2
**Rating:** 4
**Confidence:** 2

**Summary:**

HybridNorm applies QKV-Norm in the attention mechanism and Post-Norm in the feed-forward network. The paper provides theoretical analysis of gradient flow benefits and validates the approach through experiments on both dense and MoE models.

**Questions:**

- The authors claim that HybridNorm improves the stability and performance of deep transformers. However, the evaluation is limited to models with up to 16 layers, which are relatively shallow compared to mainstream LLMs—such as the 28-layer 1.5B DeepSeek R1. Can the authors provide evidence that the observed stability gains generalize to deeper architectures commonly used in practice?
- What is the computational overhead introduced by HybridNorm in terms of latency and memory consumption? How does this overhead scale with model size?
- Table 3 shows Pre-Norm and HybridNorm have different sensitivities to initialization methods. What drives this divergence, and can the author provide an intuitive explanation?

**Ethical Concerns:**

["NO or VERY MINOR ethics concerns only"]

**Final Justification:**

The rebuttal addressed my concerns well.

**Limitations:**

While HybridNorm demonstrates improvements over Pre-Norm and Post-Norm under controlled experimental settings, the lack of direct comparisons with widely recognized foundation models (e.g., LLaMA 3.2, to which the authors’ tested architecture is closely related) limits the reader’s ability to contextualize the magnitude of these gains.

For example, on the ARC-C benchmark, the 1.2B dense model with HybridNorm achieves a score of 37.06, which is lower than the Pre-Norm baseline at 39.2. Although the average score across benchmarks increases modestly from 62.99 (Pre-Norm) to 64.15 (HybridNorm∗), the relative improvements remain small. In contrast, mainstream foundation models such as Qwen2-1.5B and its enhanced variant Qwen2.5-1.5B achieve 43.7 and 54.7 on ARC-C, respectively—with the latter improvement largely attributed to higher-quality training data rather than architectural changes.

Compared to such data-driven advancements, the performance gains offered by HybridNorm appear marginal. Without contextual baselines and further explanation, it remains difficult to assess whether the improvements are meaningful or competitive relative to more influential factors such as data quality or model scale.

**Paper Formatting Concerns:**

No paper formatting concern.

**Quality:**

3

**Strengths And Weaknesses:**

- Theoretical analysis and empirical results offer clear insights into gradient stabilization.
- The paper lacks measurements of training and inference throughput, as well as memory overhead, associated with the added normalization steps. This makes it difficult to assess the practical efficiency of the proposed method.
- Experimental comparison limited to Pre-Norm, Post-Norm, and Mix-LN, advanced schemes like DeepNorm and Sandwich-LN are mentioned but not evaluated.

---

> ### Author Rebuttal · Authors · 2025-07-31
>
> Thank you for your thoughtful feedback. We appreciate that Reviewer zMv1 values our theoretical analysis. The concerns raised are mostly due to misunderstandings. We have addressed all concerns below.
>
> ---
>
> Q1. What is the computational overhead introduced by HybridNorm? How does this overhead scale with model size?
>
> A1. **The direct contribution of RMSNorm to the overall parameter count and computational cost of a large Transformer is, in fact, negligible**. Below, we provide a detailed analysis to support this claim.
>
> For simplicity, we consider only the MHA and the SwiGLU activation function, excluding the Embedding and Output layers. Suppose the hidden dimension is $d$, the intermediate FFN size is $\frac{8}{3}d$, the number of layers is $L$, and sequence length is $s$. As shown in the table below, RMSNorm constitutes only a negligible fraction of the actual runtime and memory consumption in Transformer under both Pre-Norm and HybridNorm configurations. And the ratio decreases as the model size increases. Moreover, when employing GQA, the relative overhead of RMSNorm in HybridNorm is further diminished.
>
> |  | Architecture | RMSNorm  | Main Transformer Components (MHA & FFN) | Ratio (RMSNorm / Total) | Ratio (for d=s=4096) |
> | --- | --- | --- | --- | --- | --- |
> | Parameter | Pre-Norm | $2dL$ | $12d^2L$  | $\frac{2dL}{12d^2L + 2dL} \approx \frac{1}{6d}$ | ~0.0041% |
> |  | HybridNorm | $4dL$ | $12d^2L$   | $\frac{4dL}{12d^2L + 4dL} \approx \frac{1}{3d}$ | ~0.0081% |
> | Computation (FLOPs) | Pre-Norm | $8sdL$ | $(24sd^2 + 4s^2d)L$ | $\frac{8sdL}{(24sd^2 + 4s^2d)L} = \frac{2}{6d + s}$ | ~0.0068% |
> |  | HybridNorm | $16sdL$ | $(24sd^2 + 4s^2d)L$ | $\frac{16sdL}{(24sd^2 + 4s^2d)L} = \frac{4}{6d + s}$ | ~0.0137% |
>
> **Parameters.** Each RMSNorm layer introduces $d$ parameters. In a Pre-Norm architecture, there are two RMSNorm layers per Transformer layer, resulting in a total of $2dL$ parameters. In HybridNorm, there are four RMSNorm layers per Transformer layer, yielding $4dL$ parameters in total. Each attention block has four weight matrices (for Q, K, V, and O projections), thereby contributing $4d^2L$ parameters in total for the MHA component. Similarly, the FFN component introduces an additional $8d^2L$ parameters.
>
> **Computation.**  The FLOPs for RMSNorm to process a single token vector of dimension $d$ is $(4d + 4)$. Since the constant term becomes negligible for any reasonably large $d$, this can be simplified to $4d$. Accordingly, for Pre-Norm, the $2L$ RMSNorm operations incur a total cost of approximately $8sdL$, whereas HybridNorm incurs $16sdL$. In practice, however, the dominant computational overhead in a Transformer stems from matrix-vector multiplications, primarily within MHA and FFN modules. Specifically, the FLOPs for MHA amount to $(8sd^2 + 4s^2d)L$, while the FLOPs for FFN total $16sd^2L$.
>
> ---
>
> Q2. Advanced schemes like DeepNorm and Sandwich-LN are not evaluated.
>
> A2. We have extended Table 4 to include DeepNorm, Sandwich-LN, and OutputNorm (OLMo-2), as shown below.
>
> We observe that **HybridNorm consistently outperforms all other normalization strategies**, including the aforementioned advanced methods, across all metrics. This highlights the strength of our approach in both generative perplexity tasks and HellaSwag.
>
> | **Method** | **Loss ↓** | **C4 PPL ↓** | **Pile PPL ↓** | **HellaSwag ↑** |
> | --- | --- | --- | --- | --- |
> | Post-Norm | 2.760 | 20.43 | 10.57 | 51.20 |
> | Pre-Norm | 2.751 | 20.30 | 10.48 | 51.97 |
> | Mix-LN | 2.760 | 20.43 | 10.56 | 51.29 |
> | DeepNorm | 2.746 | 20.34 | 10.48 | 52.11 |
> | Sandwich-LN | 2.751 | 20.45 | 10.58 | 52.07 |
> | OutputNorm (OLMo-2) | 2.750 | 20.34 | 10.44 | 52.82 |
> | **HybridNorm** | **2.737** | **20.00** | **10.29** | **53.35** |
> | **HybridNorm*** | **2.731** | **19.85** | **10.25** | **53.36** |
>
> ---
>
> Q3.  Provide evidence that the observed stability gains generalize to deeper architectures.
>
> A3. In fact, we have already conducted experiments on deeper models in **Section 5.5 (Deeper Models)**, where we evaluate HybridNorm on Transformers with depths ranging from 16 to **29 layers** under a comparable parameter budget. As shown in **Table 5**, both HybridNorm and HybridNorm* consistently outperform Pre-Norm and Post-Norm across all depths. Notably, **Post-Norm fails to converge at 29 layers**, while our method remains stable and achieves the best performance—demonstrating its superior scalability and training stability in deeper configurations.
>
> To further validate the generalization of our method to large-scale, practically relevant models, we now provide **additional results on a 7B dense model with 32 layers**, closely following the configuration in [2].  It is important to note that the depth of our model is aligned with that of LLaMA3 and is comparable to the depth of other mainstream open-source models. Therefore, our model should not be considered shallow by any means. The results are summarized below.
>
> Training and Validation Performance:
>
> |  | Training Loss **↓** | C4 PPL **↓** | Books PPL **↓** | Common Crawl PPL **↓** | peS2o PPL **↓** | Reddit PPL **↓** | Stack PPL **↓** | Wiki PPL **↓** | Pile PPL **↓** | Wikitext 103 PPL **↓** |
> | --- | --- | --- | --- | --- | --- | --- | --- | --- | --- | --- |
> | Pre-Norm | 2.469 | 15.32 | 13.37 | 17.10 | 8.07 | 20.31 | 3.57 | 9.96 | 7.85 | 10.09 |
> | HybridNorm* | **2.430** | **14.83** | **12.77** | **16.77** | **7.67** | **19.65** | **3.40** | **9.34** | **7.81** | **9.16** |
>
> Downstream Task Performance:
>
> |  | BasicArithmetic **↑** | HellaSwag **↑** | SciQ **↑** | ARC-C **↑** | ARC-E **↑** | PIQA **↑** | Openbook QA **↑** | COPA **↑** | WinoGrande **↑** | BoolQ **↑** | Avg. **↑** |
> | --- | --- | --- | --- | --- | --- | --- | --- | --- | --- | --- | --- |
> | Pre-Norm | 43.50 | 69.03 | 46.57 | 41.47 | 74.95 | 76.71 | 39.40 | 84.00 | 63.00 | 67.43 | 60.61 |
> | HybridNorm* | **50.67** | **70.77** | **47.44** | **43.82** | **75.82** | **78.93** | **43.77** | **86.01** | **63.32** | **70.06** | **63.06** |
>
> ---
>
> Q4. Table 3 shows Pre-Norm and HybridNorm have different sensitivities to initialization methods. What drives this divergence, and can the author provide an intuitive explanation?
>
> A4. For this issue, we attempt to provide a preliminary explanation here. In the **Pre-Norm** setup, the hidden state update follows the form $h_{t+1} = h_t + f(h_t)$. As the network depth increases, the norm of $h_t$ tends to grow, which leads to the residual branch $h_t$ increasingly dominating over the transformed signal $f(h_t)$.  In contrast, **HybridNorm** applies Post-Norm specifically to FFN, which helps to keep the norm of $h_t$ relatively stable across layers. As a result, the balance between $h_t$ and $f(h_t)$ remains more consistent during training. When using Megatron initialization, which scales down $f(h_t)$ by a factor of $1/\sqrt{2L}$, the relative contribution of $f(h_t)$ diminishes. This shifts the update dynamics closer to that of Pre-Norm, where $h_t$ dominates. As a result, HybridNorm performs better when combined with Megatron initialization.
>
> ---
>
> Q5. Limitation concerning the absence of contextual baselines.
>
> A5. Below, we provide clarifications and additional context regarding the positioning and contributions of our work.
>
> **Comparison with widely recognized foundation models.** We agree that direct comparisons with widely recognized foundation models (e.g., LLaMA 3.2, Qwen2.5) would offer valuable context. However, we would like to clarify that such comparisons are not straightforward due to substantial differences in training setups—**most notably, the total number of training tokens**. Our HybridNorm models were trained on only 1T tokens, while models like LLaMA 3.2 use up to 15T tokens, often combined with significantly higher-quality data, longer training schedules, and proprietary optimization pipelines. Under such vastly different conditions, direct performance comparisons would not be meaningful and could obscure the architectural contribution of HybridNorm. Instead, we chose to perform controlled, apples-to-apples comparisons between Pre-Norm and HybridNorm **under identical architecture, model size, token budget, and training setup**, ensuring that any observed improvements can be directly attributed to the normalization strategy. **Notably, the Pre-Norm baseline corresponds to the configuration used in LLaMA 3.2.** Since LLaMA 3.2 was trained on proprietary data, we conducted our comparisons using the same data and token budget across all settings to ensure fairness.
>
> **Orthogonality to data and scale improvements.** HybridNorm complements efforts to scale model size and data. As an architectural innovation, it can be seamlessly applied to any Transformer, regardless of scale or training corpus. Our extended experiments show **consistent performance gains even under longer training**, demonstrating strong scalability and potential for further benefits in larger, data-rich settings. Scaling laws suggest that modest performance gains often demand massive data increases—an expensive and unsustainable path. In contrast, **HybridNorm offers meaningful improvements with negligible training overhead**, enabling **faster convergence and reduced compute cost**. Given the limits of high-quality data, architectural enhancements like HybridNorm that **shift the slope or intercept of the scaling law** provide a valuable and efficient alternative.
>
> **Contribution and research focus.** Our focus is on understanding and improving the architectural design of normalization in transformers, an area that remains fundamental to the training dynamics, stability, and efficiency of large-scale models. While data and scale are undeniably critical, structure remains the foundation of model design, and our work contributes toward a better understanding of this foundational aspect.
>
> ---
>
> Hope our explanations above could ease your concerns. Please feel free to let us know if further clarification is needed.

---

> > ### Author Response · Authors · 2025-08-07
> >
> > Dear Reviewer, we sincerely appreciate your time and thoughtful feedback.
> >
> > Given the limited discussion period, we would be grateful for any additional comments or confirmation on whether our rebuttal has sufficiently addressed your concerns.

---

> > ### Comment · Reviewer_zMv1 · 2025-08-07
> >
> > My concerns and questions have been well addressed, and I will raise the rating accordingly.

---

### Official Review · Reviewer_Cq1L · 2025-07-03

**Clarity:** 3
**Significance:** 4
**Originality:** 2
**Rating:** 4
**Confidence:** 4

**Summary:**

This paper introduces a novel normalization technique that integrates post-normalization with QKV normalization to enhance the stability of transformer training. The authors demonstrate consistent performance improvements on dense LLMs (ranging from 150M to 1B parameters) and MoE models. Furthermore, the paper provides a theoretical analysis comparing the proposed solution with existing PreNorm and PostNorm approaches.

**Questions:**

Questions:
* No questions regarding the technical aspects, as they are straightforward.
* Have the authors explored scaling the model beyond 7B parameters?

**Ethical Concerns:**

["NO or VERY MINOR ethics concerns only"]

**Limitations:**

yes

**Quality:**

3

**Strengths And Weaknesses:**

Strengths:
* The development of improved normalization techniques is crucial for scaling base models, addressing a significant concern regarding training stability.
* The proposed solution is not only empirically validated but also theoretically supported.
* The experimental setup is extensive, although an evaluation on larger model scales (e.g., > 7B) would be beneficial.

Weaknesses:
* The method, primarily an exploration of normalization placement, has limited novelty.
* Evaluating the solution solely on models under 1B parameters may not be sufficient to assess its scalability and effectiveness in real-world LLM training scenarios.

---

> ### Author Rebuttal · Authors · 2025-07-31
>
> Thanks for your careful reading and insightful comments. We appreciate that Reviewer Cq1L recognizes the importance of improved normalization techniques for stable transformer training and acknowledges the empirical strength and theoretical soundness of our proposed method. We address your concerns below.
>
> ---
>
> Q1. The method, primarily an exploration of normalization placement, has limited novelty.
>
> A1. First and foremost, we would like to emphasize that our method, HybridNorm, is a **simple yet effective** approach to enhancing both model performance and training stability. The core idea is to integrate Pre-Norm and Post-Norm within each Transformer layer, to balance training stability and model performance. Moreover, HybridNorm*  introduces a specialized treatment for a specific layer.HybridNorm adopts a hybrid strategy within each transformer block, leading to more stable gradient propagation during backpropagation and effectively mitigating both gradient explosion and vanishing. Moreover, our method also achieves the best performance among all evaluated approaches.
>
> Building on the foundation of the scaling laws, even seemingly minor modifications—such as changes to model architecture, hyperparameter configurations, initialization schemes, or training protocols—can lead to substantial shifts in performance, particularly in the context of large-scale LLMs. Techniques like GQA and MQA exemplify this phenomenon. Although our proposed changes may appear modest, **they induce significant alterations in training dynamics**, especially under a large training budget. Furthermore, we observe a clear improvement in convergence speedup. Extensive empirical studies and ablation analyses further validate the simplicity and effectiveness of our approach.
>
> Therefore, our contribution goes beyond a simple combination of existing components and offers a non-trivial advancement over prior approaches.
>
> ---
>
> Q2. Have the authors explored scaling the model beyond 7B parameters?
>
> A2. The models we have trained so far scale up to a maximum size of 7B parameters. In fact, as described in Section 5.2 of the paper, we have already trained a 7B-parameter MoE model with 1B active parameters using 500B training tokens, and demonstrated that HybridNorm* achieves superior performance. Additionally, we have included experiments with a 7B dense model for further comparison.
>
> **New experiment in 7B dense model.** The experimental setup primarily follows the 7B model configuration outlined in [2] and the number of training tokens is 150B. The first table below presents the training loss and validation perplexity (PPL), while the second table reports the downstream evaluation metrics.
>
> |  | Training Loss **↓** | C4 PPL **↓** | Books PPL **↓** | Common Crawl PPL **↓** | peS2o PPL **↓** | Reddit PPL **↓** | Stack PPL **↓** | Wiki PPL **↓** | Pile PPL **↓** | Wikitext 103 PPL **↓** |
> | --- | --- | --- | --- | --- | --- | --- | --- | --- | --- | --- |
> | Pre-Norm | 2.469 | 15.32 | 13.37 | 17.10 | 8.07 | 20.31 | 3.57 | 9.96 | 7.85 | 10.09 |
> | HybridNorm* | **2.430** | **14.83** | **12.77** | **16.77** | **7.67** | **19.65** | **3.40** | **9.34** | **7.81** | **9.16** |
>
> |  | BasicArithmetic **↑** | HellaSwag **↑** | SciQ **↑** | ARC-C **↑** | ARC-E **↑** | PIQA **↑** | Openbook QA **↑** | COPA **↑** | WinoGrande **↑** | BoolQ **↑** | Avg. **↑** |
> | --- | --- | --- | --- | --- | --- | --- | --- | --- | --- | --- | --- |
> | Pre-Norm | 43.50 | 69.03 | 46.57 | 41.47 | 74.95 | 76.71 | 39.40 | 84.00 | 63.00 | 67.43 | 60.61 |
> | HybridNorm* | **50.67** | **70.77** | **47.44** | **43.82** | **75.82** | **78.93** | **43.77** | **86.01** | **63.32** | **70.06** | **63.06** |
>
> The experimental results demonstrate the clear superiority of **HybridNorm*** over the traditional **Pre-Norm** approach in the 7B model. Firstly, HybridNorm* achieves a lower training loss (2.430 vs. 2.469), indicating more efficient optimization during training. This improvement in training dynamics translates into consistently better performance across a range of language modeling benchmarks. Specifically, HybridNorm* yields lower perplexity scores on all evaluated datasets, including C4, Books, Common Crawl, Wiki, and Wikitext 103. For instance, perplexity on the C4 dataset drops from 15.32 to 14.83, suggesting stronger generalization across both structured and unstructured corpora.
>
> Moreover, HybridNorm* shows consistent improvements on all downstream tasks, covering various domains such as arithmetic reasoning, commonsense QA, and natural language inference. Notably, performance on BasicArithmetic improves significantly from 43.50% to 50.67%. Across the full suite of tasks—including ARC, PIQA, COPA, BoolQ, and WinoGrande—HybridNorm* outperforms Pre-Norm in every case, leading to an overall increase in average accuracy from 60.61% to 63.06%.
>
> ---
>
> Please feel free to let us know if further clarification is needed.

---

> > ### Comment · Reviewer_Cq1L · 2025-08-05
> >
> > I have read the responce and will keep my positive rating.

---

### Official Review · Reviewer_gyxV · 2025-07-03

**Clarity:** 3
**Significance:** 3
**Originality:** 3
**Rating:** 5
**Confidence:** 3

**Summary:**

This paper proposes the HybridNorm method by combining the advantages of Pre-Norm and Post-Norm, it innovatively designs the normalization strategies in Transformer training, especially normalizing QKV in attention mechanism, and adopting Post-Norm in FFN, which in theory helps improve training stability and model performance. The author not only gave theoretical analysis, but also proved the effectiveness of HybridNorm through a large number of experiments, showing the application potential of this method in actual large-scale model training.

**Questions:**

1.From the overall perspective of the paper, the paper mainly focuses on performance improvement and training stability, but there is less interpretability analysis of the internal mechanism of HybridNorm. Why not you use some feature maps of different layers and experiments with heat maps to further verify the effectiveness of the method?
2.The method proposed in this paper is carried out from the model with 151M to 7B parameters, but the model with a larger parameter quantity has not been fully experimentally verified. Why?

**Ethical Concerns:**

["NO or VERY MINOR ethics concerns only"]

**Final Justification:**

I would like to commend the authors for their thorough and thoughtful response to my initial review. The rebuttal satisfactorily addresses the points I raised. In particular, the detailed explanations and additional experiments/analyses provided in the appendix have effectively alleviated my concerns.

Overall, this paper presents a new normalization method with significant general applicability. The paper is well-structured. The high quality of both the research and the writing is evident.

Given the authors' successful addressing of my comments and the inherent strengths of the paper, I am now convinced of its merit and strongly recommend it for acceptance.

**Limitations:**

See the weaknesses.

**Paper Formatting Concerns:**

There are some formatting issues, such as punctuation issues in line 40 of the second page, “... in Appendix A..”.

**Quality:**

4

**Strengths And Weaknesses:**

Strengths:
The author conducted research in the normalization process of Transformer and proposed a training method that can efficiently transformer architecture. It systematically compared with existing mainstream normalization methods such as Pre-Norm, Post-Norm, QK-Norm, etc. The experimental part covers a variety of model structures and data sets, has a relatively comprehensive index, and is meticulous in the ablation experiment, which can fully verify the effectiveness of the method and the role of each component. It also displays a complete theoretical derivation process in the supplementary material, proving the effectiveness of the proposed method.

Weaknesses:
1.From the overall perspective of the article, the paper mainly focuses on performance improvement and training stability, but there is less interpretability analysis of the internal mechanism of HybridNorm, which can supplement some feature maps of different layers and experiments with heat maps to further verify the effectiveness of the method.
2.The method proposed in this paper is carried out from the model with 151M to 7B parameters, but the model with a larger parameter quantity has not been fully experimentally verified. I hope that additional experimental verification can be carried out here in the future.

---

> ### Author Rebuttal · Authors · 2025-07-31
>
> Thank you for your thoughtful feedback and constructive comments. We appreciate that Reviewer gyxV finds our method novel, our experiments well-executed, and our theoretical derivation complete. Below, we address your concerns point by point.
>
> ---
>
> Q1. From the overall perspective of the paper, the paper mainly focuses on performance improvement and training stability, but there is less interpretability analysis of the internal mechanism of HybridNorm. Why not you use some feature maps of different layers and experiments with heat maps to further verify the effectiveness of the method?
>
> A1. Thank you very much for your insightful feedback. We fully agree that a more in-depth interpretability analysis could significantly enhance the understanding of HybridNorm’s internal mechanisms. **In fact, we have taken initial steps in this direction in Appendix D.2 and D.3 of our paper, where we examine information flow through the perspectives of signal propagation and entropy dynamics during pre-training.** The visualizations in Figure 7 and Figure 8 illustrate how HybridNorm influences layerwise entropy evolution and signal behavior across the network. We greatly appreciate your suggestion to incorporate feature map and heatmap-based analyses. We see this as a promising direction and plan to further expand our interpretability efforts along these lines in future work.
>
> ---
>
> Q2. The method proposed in this paper is carried out from the model with 151M to 7B parameters, but the model with a larger parameter quantity has not been fully experimentally verified. Why?
>
> A2. In fact, as described in Section 5.2 of the paper, we have already trained a 7B-parameter MoE model with 1B active parameters using 500B training tokens, and demonstrated that HybridNorm* achieves superior performance. Additionally, we have included experiments with a 7B dense model for further comparison below.
>
> **New experiment in 7B dense model.** The experimental setup primarily follows the 7B model configuration outlined in [2] and the number of training tokens is 150B. The first table below presents the training loss and validation perplexity (PPL), while the second table reports the downstream evaluation metrics.
>
> |  | Training Loss **↓** | C4 PPL **↓** | Books PPL **↓** | Common Crawl PPL **↓** | peS2o PPL **↓** | Reddit PPL **↓** | Stack PPL **↓** | Wiki PPL **↓** | Pile PPL **↓** | Wikitext 103 PPL **↓** |
> | --- | --- | --- | --- | --- | --- | --- | --- | --- | --- | --- |
> | Pre-Norm | 2.469 | 15.32 | 13.37 | 17.10 | 8.07 | 20.31 | 3.57 | 9.96 | 7.85 | 10.09 |
> | HybridNorm* | **2.430** | **14.83** | **12.77** | **16.77** | **7.67** | **19.65** | **3.40** | **9.34** | **7.81** | **9.16** |
>
> |  | BasicArithmetic **↑** | HellaSwag **↑** | SciQ **↑** | ARC-C **↑** | ARC-E **↑** | PIQA **↑** | Openbook QA **↑** | COPA **↑** | WinoGrande **↑** | BoolQ **↑** | Avg. **↑** |
> | --- | --- | --- | --- | --- | --- | --- | --- | --- | --- | --- | --- |
> | Pre-Norm | 43.50 | 69.03 | 46.57 | 41.47 | 74.95 | 76.71 | 39.40 | 84.00 | 63.00 | 67.43 | 60.61 |
> | HybridNorm* | **50.67** | **70.77** | **47.44** | **43.82** | **75.82** | **78.93** | **43.77** | **86.01** | **63.32** | **70.06** | **63.06** |
>
> The experimental results demonstrate the clear superiority of **HybridNorm*** over the traditional **Pre-Norm** approach in the 7B model. Firstly, HybridNorm* achieves a lower training loss (2.430 vs. 2.469), indicating more efficient optimization during training. This improvement in training dynamics translates into consistently better performance across a range of language modeling benchmarks. Specifically, HybridNorm* yields lower perplexity scores on all evaluated datasets, including C4, Books, Common Crawl, Wiki, and Wikitext 103. For instance, perplexity on the C4 dataset drops from 15.32 to 14.83, suggesting stronger generalization across both structured and unstructured corpora.
>
> Moreover, HybridNorm* shows consistent improvements on all downstream tasks, covering various domains such as arithmetic reasoning, commonsense QA, and natural language inference. Notably, performance on BasicArithmetic improves significantly from 43.50% to 50.67%. Across the full suite of tasks—including ARC, PIQA, COPA, BoolQ, and WinoGrande—HybridNorm* outperforms Pre-Norm, leading to an overall increase in average accuracy from 60.61% to 63.06%.
>
> ---
>
> Thank you for pointing out our type. We have corrected it accordingly. Please feel free to let us know if further clarification is needed.

---

> > ### Author Response · Authors · 2025-08-07
> >
> > Dear Reviewer, we sincerely appreciate your time and thoughtful feedback.
> >
> > Given the limited discussion period, we would be grateful for any additional comments or confirmation on whether our rebuttal has sufficiently addressed your concerns.

---

### Official Review · Reviewer_RJ3j · 2025-07-05

**Clarity:** 3
**Significance:** 2
**Originality:** 2
**Rating:** 3
**Confidence:** 4

**Summary:**

This paper introduces HybridNorm, a normalization strategy for transformers that aims to combine the benefits of both Pre-Norm and Post-Norm approaches to address challenges in training deep transformer networks. HybridNorm specifically applies QKV normlization within the attention mechanism and Post-Norm in FFN of each block. The authors provide theoretical analysis and empirical evidence to demonstrate that HybridNorm improves gradient flow and model robustness.

**Questions:**

1. Could the authors elaborate on the specific differences in how RMSNorm and LayerNorm  might influence the theoretical gradient analysis and the empirical performance of HybridNorm?
2. Please empirically validate HybridNorm on 7B or larger models
3. Fig. 6 in [1] uses QKV-Norm to stably train a Transformer. It seems that you only changed pre-norm to post-norm. Both of the pre- and post-norm have advantages, they have their own shortcomings, too. You used post-norm, which introduces the Gradient Vanish problem mentioned in [2]

**Ethical Concerns:**

["NO or VERY MINOR ethics concerns only"]

**Final Justification:**

Adjust accordingly after the rebuttal phase.

**Limitations:**

Yes

**Quality:**

2

**Strengths And Weaknesses:**

Strength:

1. HybridNorm outperforms both Pre-Norm and Post-Norm across dense and Mixture of Experts models onmultiple benchmarks, demonstrating improved training stability and better downstream performance.
2. This paper provides theoretical insights and gradient analysis to explain why HybridNorm improves gradient flow stability and model robustness.

Weakness:
1. The paper mentions that their theoretical analysis withRMSNorm. While RMSNorm and LayerNorm are common, the effectiveness of HybridNorm might be sensitive to the specific normalization function chosen, and the theoretical guarantees might not directly translate if other normalization techniques were considered

2. The related work section mentions DeepNorm, Sandwich-LN, and OLMo-2 as normalization strategies. While the paper generally compares against Pre-Norm and Post-Norm, a more direct and in-depth empirical comparison with these advanced normalization methods could further solidify the superior performance of HybridNorm.

3. This paper provides limited technical contribution. The method is very similar to Figure 6 in [1], merely replacing pre-norm with post-norm.

---

> ### Author Rebuttal · Authors · 2025-07-31
>
> Thanks for your time and constructive comments. We appreciate that Reviewer RJ3j finds our method effective and consistent with our theoretical analysis. We address your concerns below and hope our responses adequately clarify your questions.
>
> ---
>
> Q1.  Could the authors elaborate on the specific differences in how RMSNorm and LayerNorm might influence the theoretical gradient analysis and the empirical performance of HybridNorm?
>
> A1. In the following, we demonstrate that RMSNorm and LayerNorm exhibit no fundamental differences, both in theoretical analysis and empirical observations. It is worth noting that, given the vast majority of popular LLMs are based on RMSNorm, our experiments and conclusions are broadly applicable to standard LLM architectures.
>
> **On the theoretical side**, we adopt the notation introduced in Appendix B of the paper. Suppose the input is given by $X \in \mathbb{R}^{s \times d}$. Let $P= I - \frac{1}{d}1_d1_d^{\top}$ .  Then $\mathbb{E}X = X \frac{1}{d}1_d1_d^{\top}$ and  $X-\mathbb{E}X =X - X \frac{1}{d}1_d1_d^{\top}=XP$. For simplicity, we omit the affine transformation in the normalization layers. It follows that
>
> $$
> LayerNorm(X)=\frac{X-\mathbb{E}X}{Var(X)}=\frac{X-\mathbb{E}X}{RMS(X-\mathbb{E}X)}=\frac{XP}{RMS(XP)}=RMSNorm(XP).
> $$
>
> Hence, analogous to the gradient of RMSNorm (Eq. (20) in the paper), the gradient of LayerNorm is given by
>
> $$
> \frac{\partial LayerNorm(X)}{\partial X} = \frac{\partial RMSNorm(XP)}{\partial X} = \mathrm{blockdiag}\left(\frac{\partial RMSNorm((XP)_i)}{\partial X_i}\right)=\mathrm{blockdiag}\left( \frac{\sqrt{d}}{||(XP)_i||_2}\left(I_d - \frac{(XP)_i(XP)_i^\top}{||(XP)_i||_2^2} \right)P\right).
> $$
>
> Here, $X_i$ means that the i-th column of $X$. Note that $P$ is a positive semidefinite matrix with eigenvalues bounded between 0 and 1, hence $||P||_2 = 1$. As a result, the gradient norms of LayerNorm and RMSNorm differ only by a constant factor. **This implies that the main result, Theorem 2 in the paper, also holds for LayerNorm.**
>
> **On the experimental side,** we conducted  controlled comparison using models of identical size (550M parameters), each trained on 400B tokens. Both models adopt the Pre-Norm architecture and employ the Megatron initialization scheme; the only difference lies in the normalization method—**one uses RMSNorm, while the other uses LayerNorm**. Detailed model configurations and experimental hyperparameters are provided in Table 6 and Table 7 of the paper. As shown in the table below, **the training losses of RMSNorm and LayerNorm are nearly identical**, with a marginal difference of only 0.0008. In the context of large-scale models, a loss difference smaller than 0.001 is typically considered negligible.
>
> |  | Training Loss |
> | --- | --- |
> | RMSNorm | 2.7631 |
> | LayerNorm | 2.7639 |
>
> ---
>
> Q2. A more direct and in-depth empirical comparison with these advanced normalization methods could further solidify the superior performance of HybridNorm.
>
> A2. We have extended Table 4 to include **DeepNorm**, **Sandwich-LN**, and **OutputNorm (OLMo-2** [2]**)**, as shown below. We have also clarified in the revised manuscript that **OutputNorm** refers to the normalization strategy employed in **OLMo-2**, where RMSNorm is applied to the outputs of attention and MLP sublayers.
>
> It shows that **HybridNorm consistently outperforms all other normalization strategies**, including the aforementioned advanced methods, across all metrics. This highlights the strength of our approach in both generative perplexity tasks and HellaSwag.
>
> | **Method** | **Loss ↓** | **C4 PPL ↓** | **Pile PPL ↓** | **HellaSwag ↑** |
> | --- | --- | --- | --- | --- |
> | Post-Norm | 2.760 | 20.43 | 10.57 | 51.20 |
> | Pre-Norm | 2.751 | 20.30 | 10.48 | 51.97 |
> | Mix-LN | 2.760 | 20.43 | 10.56 | 51.29 |
> | DeepNorm | 2.746 | 20.34 | 10.48 | 52.11 |
> | Sandwich-LN | 2.751 | 20.45 | 10.58 | 52.07 |
> | OutputNorm (OLMo-2) | 2.750 | 20.34 | 10.44 | 52.82 |
> | **HybridNorm** | **2.737** | **20.00** | **10.29** | **53.35** |
> | **HybridNorm*** | **2.731** | **19.85** | **10.25** | **53.36** |
>
> ---
>
> Q3. This paper provides limited technical contribution. The method is very similar to Figure 6 in [1], merely replacing pre-norm with post-norm.. Both of the pre- and post-norm have advantages, they have their own shortcomings, too. You used post-norm, which introduces the Gradient Vanish problem.
>
> A3. First and foremost, we would like to emphasize that our method, HybridNorm, is a **simple yet effective** approach to enhancing both model performance and training stability. The core idea is to integrate Pre-Norm and Post-Norm within each Transformer layer, to balance training stability and model performance. Moreover, HybridNorm*  introduces a specialized treatment for a specific layer.
>
> The primary difference between our method and the approach shown in Fig. 6 of [1] (which we refer to as QKV-Pre) lies in the integration of different normalization types within each transformer block. The layer-wise gradient norm of QKV-Pre, similar to Pre-Norm illustrated in Figure 2, tends to exhibit gradient explosion in deeper models, while Post-Norm suffers from vanishing gradients. **In contrast, HybridNorm adopts a hybrid strategy within each transformer block, leading to more stable gradient propagation during backpropagation and effectively mitigating both gradient explosion and vanishing.** Moreover, our method achieves the best performance. We also provides theoretical insights for QKV-Norm.
>
> Building on the foundation of the scaling laws, even seemingly minor modifications—such as changes to model architecture, hyperparameter configurations, initialization schemes, or training protocols—can lead to substantial shifts in performance, particularly in the context of large-scale LLMs. Techniques like GQA and MQA exemplify this phenomenon. Although our proposed changes may appear modest, **they induce significant alterations in training dynamics**, especially under a large training budget. Furthermore, we observe a clear improvement in convergence speedup. Extensive empirical studies and ablation analyses further validate the simplicity and effectiveness of our approach.
>
> Therefore, our contribution goes beyond a simple combination of existing components and offers a non-trivial advancement over prior approaches.
>
> ---
>
> Q4.  Please empirically validate HybridNorm on 7B or larger models.
>
> A4. In fact, as described in Section 5.2 of the paper, we have already trained a 7B-parameter MoE model with 1B active parameters using 500B training tokens, and demonstrated that HybridNorm* achieves superior performance. Additionally, we have included experiments with a 7B dense model for further comparison.
>
> **New experiment in 7B dense model.** The experimental setup primarily follows the 7B model configuration outlined in [2] and the number of training tokens is 150B. The first table below presents the training loss and validation perplexity (PPL), while the second table reports the downstream evaluation metrics.
>
> |  | Training Loss **↓** | C4 PPL **↓** | Books PPL **↓** | Common Crawl PPL **↓** | peS2o PPL **↓** | Reddit PPL **↓** | Stack PPL **↓** | Wiki PPL **↓** | Pile PPL **↓** | Wikitext 103 PPL **↓** |
> | --- | --- | --- | --- | --- | --- | --- | --- | --- | --- | --- |
> | Pre-Norm | 2.469 | 15.32 | 13.37 | 17.10 | 8.07 | 20.31 | 3.57 | 9.96 | 7.85 | 10.09 |
> | HybridNorm* | **2.430** | **14.83** | **12.77** | **16.77** | **7.67** | **19.65** | **3.40** | **9.34** | **7.81** | **9.16** |
>
> |  | BasicArithmetic **↑** | HellaSwag **↑** | SciQ **↑** | ARC-C **↑** | ARC-E **↑** | PIQA **↑** | Openbook QA **↑** | COPA **↑** | WinoGrande **↑** | BoolQ **↑** | Avg. **↑** |
> | --- | --- | --- | --- | --- | --- | --- | --- | --- | --- | --- | --- |
> | Pre-Norm | 43.50 | 69.03 | 46.57 | 41.47 | 74.95 | 76.71 | 39.40 | 84.00 | 63.00 | 67.43 | 60.61 |
> | HybridNorm* | **50.67** | **70.77** | **47.44** | **43.82** | **75.82** | **78.93** | **43.77** | **86.01** | **63.32** | **70.06** | **63.06** |
>
> The experimental results demonstrate the clear superiority of **HybridNorm*** over the traditional **Pre-Norm** approach in the 7B model. Firstly, HybridNorm* achieves a lower training loss (2.430 vs. 2.469), indicating more efficient optimization during training. This improvement in training dynamics translates into consistently better performance across a range of language modeling benchmarks. Specifically, HybridNorm* yields lower perplexity scores on all evaluated datasets, including C4, Books, Common Crawl, Wiki, and Wikitext 103. For instance, perplexity on the C4 dataset drops from 15.32 to 14.83, suggesting stronger generalization across both structured and unstructured corpora.
>
> Moreover, HybridNorm* shows consistent improvements on all downstream tasks, covering various domains such as arithmetic reasoning, commonsense QA, and natural language inference. Notably, performance on BasicArithmetic improves significantly from 43.50% to 50.67%. Across the full suite of tasks—including ARC, PIQA, COPA, BoolQ, and WinoGrande—HybridNorm* outperforms Pre-Norm in every case, leading to an overall increase in average accuracy from 60.61% to 63.06%.
>
> ---
>
> Hope our explanations address your concerns. Please feel free to let us know if further clarification is needed.
>
> [1] Rybakov, et at. Methods of improving LLM training stability.  arXiv preprint arXiv: 2410.16682
>
> [2] OLMo Team, et al. 2OLMo 2 Furious. arXiv preprint arXiv:2501.00656

---

> > ### Comment · Reviewer_RJ3j · 2025-08-05
> >
> > Although some of the concerns remain, I would like to thank the authors for the efforts during the rebuttal phase. I adjust the rating accordingly given the response and other reviews.

---

### Author Response · Authors · 2025-08-09
**General Response**

Dear Reviewers and Area Chairs,

We are truly grateful for the thorough insights and constructive feedback offered by the reviewers during the review process, as well as the valuable guidance from the Area Chairs. As we reach the end of the rebuttal phase, we would like to highlight the key contributions of our manuscript, "HybridNorm: Towards Stable and Efficient Transformer Training via Hybrid Normalization", and share our reflections on the discussions surrounding the feedback we received.

**Main Contributions**

We propose HybridNorm, a **simple yet effective** hybrid normalization strategy that seamlessly integrates the advantages of both Pre-Norm and Post-Norm. Our model architecture highlights the following key advantages:

a) **Theoretically superior gradient flow and model robustness**: We present rigorous theoretical and empirical analyses of HybridNorm to validate its training stability and model performance, underscoring the method's effectiveness in mitigating core challenges inherent in deep transformer architectures.

b) **Almost a free lunch**: the overhead and memory footprint of HybridNorm are negligible, yet it ensures robust convergence speed during training (about 1.4x speedup) and superior final performance across a wide range of metrics.

c) **Extensive validation on large-scale models**: we empirically validate the effectiveness of our approach on 1B-Dense, 7B-Dense models and 1B-7B MoE models, while also comparing it to other relevant baselines such as Pre-Norm (with LLaMA-3.2 architecture), Post-Norm, DeepNorm, Sandwich-LN, and Mix-LN, thoroughly demonstrating the method's efficacy.

In response to the constructive feedback from the reviewers, we have made several significant additions:

**Addressing Common Reviewer Comments:**

1. **Expanded Theoretical Clarifications**: Based on the reviewers' comments, we have further clarified and expanded the theoretical contributions of our approach. The refined explanations have addressed the remaining questions regarding the proof details.
2. **Enhanced Empirical Analysis**: For more empirical analyse, we have included feature visualizations, heatmaps, and comprehensive analyses in the appendix to provide deeper insights into the effectiveness of our method. We welcome the opportunity to discuss these supplementary results in greater detail.
3. **Expanded (Larger, Deeper, More Diverse) Experimental Evaluations**: Following the reviewers' recommendations, we have conducted extensive additional experiments, including comparisons with 7B-Dense model, and various normalization baselines (e.g., Sandwich-LN, DeepNorm).  And the experiment of deeper models, already reported in Section 5.5, further validate the scalability and generalizability of our proposed HybridNorm approach.
4. **Clarifying Contributions and Research Focus**: Our work focuses on advancing the architectural design of normalization in transformers, **a fundamental aspect of model training dynamics, stability, and efficiency**. While data and scale are critical, we argue that architectural enhancements like HybridNorm can provide **meaningful and cost-effective performance improvements**, complementing efforts to scale model size and data. HybridNorm can be seamlessly integrated into any transformer architecture, offering consistent gains even under longer training, thereby shifting the slope or intercept of the scaling law in a valuable and efficient manner.
5. **Computational Overhead**: As noted in the review response for zMv1, the additional parameters and computational cost introduced by HybridNorm are negligible (**0.004% and 0.0069%**, respectively), further demonstrating the efficiency of our approach.

We believe the revisions and clarifications provided have thoroughly addressed the reviewers’ key concerns. We are confident that the updated manuscript now presents our theoretical and empirical contributions with greater clarity and impact.

Once again, we extend our sincere gratitude to the reviewers and area chairs for their constructive feedback and guidance. Your invaluable input has been instrumental in refining and strengthening our research.

Best regards,

The Authors of Submission #22300

---

### Decision · Program_Chairs · 2025-09-17

**Decision:**

Accept (poster)

**Comment:**

Pre-norm and post-norm are two norm approaches used in transformer blocks, each with a different set of benefits. This paper proposes a way to combine them by using pre-norm in the QKV transforms of the attention mechanism and post-norm in the FFN section. The authors show theory and experiments to show that this results in better gradient flow and robustness.

Strengths
- Outperforms pre and post norm, and QK norm on dense and MoE models on multiple benchmarks. This directly improves the transformer architecture and is therefore, highly impactful.
- All reviewers appreciate the theoretical insights from this paper.

Weaknesses
- Many weaknesses have been reasonably addressed by the authors in the rebuttal
    - New comparisons provided with DeepNorm, Sandwich LN and Olmo-2 norm
    - Testing on very deep architectures
    - Provided details of only negligible computational overheads
    - Figures illustrating how HybridNorm influences layerwise entropy evolution and signal behavior across the network.

For this paper, I am waiting for reviewers to provide their final justification. R1 seems to not have updated their score after rebuttal but have indicated that they are willing to do so.  Similarly, R4 has acknowledged that all their concerns have been addressed but have not updated their score. Once those scores are in, I would like to recommend weakly accepting this paper.